# PERTURBED FLOW MATCHING FOR STRUCTURE-BASED DRUG DESIGN

## ABSTRACT

Generating 3D molecules that bind to specific protein targets via generative models has shown great promise in structure-based drug design. Recent diffusion-based approaches are constrained by marginals known in closed form and lack a design of the conditional probability path, which may hinder improved molecular generation. To address this issue, we introduce a flow-matching–based method—Perturbed Flow Matching (**PFM**)—which introduces a unique *perturbed conditional probability path design* that incorporates pocket binding site information and atom type–coordinate coupled information to enhance molecular generation performance. Experiments on the CrossDocked2020 dataset show that our model generates molecules with competitive 3D structures and state-of-the-art (SOTA) binding affinities toward protein targets, achieving an average score of -7.83. Validation on broader molecular datasets further confirms the consistent effectiveness of the proposed method. The code is available at
https://anonymous.4open.science/r/pfm_ev_rv.

## 1 INTRODUCTION

Structure-based drug design (SBDD) (Anderson, 2003) is one of the key approaches in modern drug discovery. It aims to design 3D ligand molecules with specific properties, such as high binding affinity, conditioned on a target binding site. With the development of deep learning, many generative methods have emerged to address this task. Among these, autoregressive models (Luo et al., 2021; Liu et al., 2022a; Peng et al., 2022), which iteratively construct molecules by adding atoms and bonds, have made notable advancements. Although (Zhang et al., 2023) uses fragment-based strategies to generate higher-quality molecules, this kind of method is still constrained by generation order. Recently, diffusion models (Sohl-Dickstein et al., 2015; Ho et al., 2020; Song et al., 2020) based on stochastic dynamics, have demonstrated remarkable performance. The main idea of these methods lies in denoising by inferring the reverse of the diffusion process through models. Guan et al. (2023a) introduced diffusion models into SBDD, and subsequent work often augments models with prior knowledge (Guan et al., 2023b; Huang et al., 2024).

However, exploration of the data and the generative process itself has been limited, so key information—specifically a type–coordinate coupling within molecules, as well as intermolecular interactions—can be partially omitted when constructing the conditional path of generative process. Recent studies (Song et al., 2024; Tong et al., 2024) show that the choice of conditional probability paths materially affects generative outcomes. Motivated by these observations and IPDiff (Huang et al., 2024), we propose a Flow Matching (FM) method (Lipman et al., 2022; Tong et al., 2023; Chen & Lipman, 2024) with a perturbed conditional probability path that incorporates pocket binding-site information and atom type–coordinate coupling, termed Perturbed Flow Matching (PFM). PFM is trained with a designed full-atom multi-stage equivariant network, and main experiments on the CrossDocked2020 dataset (Francoeur et al., 2020) show the effectiveness. To further improve molecular generation performance in SBDD, we introduce a loss that accounts for molecular geometry and steric clash, and also explore the use of classifier guidance and classifier-free guidance (Lipman et al., 2024). Moreover, experiments on QM9 (Ruddigkeit et al., 2012) and BindingMOAD (Hu et al., 2005) datasets provide more extensive evidence for the effectiveness of the perturbed conditional probability path.

Our contributions are as follows:

1. We introduce Perturbed Flow Matching (PFM) for molecular generation, conditioned on a target binding site.

2. A Full-atom Multi-stage Equivariant Network is proposed to enable flexible training of PFM.

3. Different guidance methods improve the quality of generated molecules.

4. Extensive validation across multiple molecular datasets demonstrates the effectiveness of the perturbed conditional probability path design.

## 2 RELATED WORK

**Structure-Based Drug Design** SBDD aims to generate 3D molecules with desirable properties (e.g., high affinity) conditioned on a target binding site. Skalic et al. (2019) and Xu et al. (2021) proposed methods for generating SMILES of molecules from given proteins. Luo et al. (2021), Liu et al. (2022a), and Peng et al. (2022) developed approaches for generating molecules in 3D Euclidean space using autoregressive techniques. Recently, the development of diffusion models has provided new solutions for SBDD tasks. Guan et al. (2023a), Lin et al. (2025), and Schneuing et al. (2024) decomposed the molecular diffusion process into continuous diffusion of atom coordinates and discrete diffusion of atom types. After generating coordinates and types, they used tools (O'Boyle et al., 2011) to restore bonds and reconstruct the complete molecular structure. A key distinction between Guan et al. (2023a) and Lin et al. (2025) lies in their variance schedules. Schneuing et al. (2024) incorporates additional considerations such as off-the-shelf property optimization, explicit negative design, and partial molecular design with inpainting. Guan et al. (2023b) and Huang et al. (2024) take prior knowledge into account. The former utilizes AlphaSpace2 (Katigbak et al., 2020) and BRICS (Degen et al., 2008) to extract the target protein subpockets and decompose molecules into fragments, while the latter introduces prior perturbations to distinguish differences among pocket binding sites by incorporating embeddings from models trained for binding affinity prediction on external datasets (Liu et al., 2015). Most non-autoregressive methods do not explicitly design the generative process, and the type–coordinate–pocket coupling is accessed only through marginal paths and left to be learned by the model. We instead directly incorporate this information into the conditional probability path to induce perturbations, which intuitively yields a more informative and discriminative conditional process and improves model performance.

**Flow Matching and Applications** Flow-based generative models (Chen et al., 2018; Grathwohl et al., 2018; Hoogeboom et al., 2021) faced training challenges due to the computational expense of integrating ODEs in simulation-based training objectives. Recent advancements in simulation-free Flow Matching (Lipman et al., 2022) methods have overcome this limitation. Current FM approaches focus on exploring probability paths (Tong et al., 2023; 2024) and extending the theory to handle general manifolds (Chen & Lipman, 2024). Moreover, Discrete Flow Matching (Campbell et al., 2024; Gat et al., 2024) has emerged, enabling the generation of complex data. In the field of molecular generation, Song et al. (2024) proposed an equivariant flow matching method for unconditional molecular generation, and Nori & Jin (2024) applied FM methods to RNA generation. Campbell et al. (2024) developed a Discrete Flow Matching approach based on Continuous Time Markov Chains (CTMCs) for protein sequence and structure generation, and Li et al. (2024)'s continuous processing of amino acid types has shown promising results. In this work, for simplicity, we adopt the Gaussian path from Tong et al. (2023) and Li et al. (2024). Building on this path, we perturb it by incorporating estimates of atom types or coordinates from pretrained predictors, with a time-dependent coefficient that preserves the right marginals at $t = 0$ and $t = 1$.

## 3 METHOD

In this section, we detail PFM. Sec.3.1 reviews flow matching. Sec.3.2 formulates the problem and specifies the basic conditional probability path. Sec.3.3 introduces its perturbation. Sec.3.4 presents a loss that jointly models the flow-matching dynamics and intra-/intermolecular interactions, and outlines the practical procedure for obtaining perturbations. Sec.3.5 describes the Full-Atom Multi-Stage Equivariant Network used for training and inference. Sec.3.6 details guidance for generation.

### 3.1 PRELIMINARIES ON FLOW MATCHING

Consider a predefined marginal probability path (or target flow) $p_t : [0, 1] \times \mathbb{R}^d \to \mathbb{R}^+$, which interpolates between the prior distribution $p_0$ and the data distribution $p_1 = q$. The flow $p_t$ is generated by a smooth vector field $u_t^\circ(x) : [0, 1] \times \mathbb{R}^d \to \mathbb{R}^d$, where $u_t^\circ$ defines the deterministic process $dx = u_t^\circ(x)\,dt$ (with *continuity equation* $\partial_t p_t = -\nabla \cdot (p_t u_t^\circ)$ holding in this case (Villani et al., 2009)). Approximating $u_t^\circ(x)$ using a parameterized vector field $v_t^\theta(x)$ enables the learning of a flow that matches $p_t$, while the the loss function $\mathcal{L}_{FM}(\theta) = \mathbb{E}_{t,p_t(x)}\|v_t^\theta(x) - u_t^\circ(x)\|_2^2$ is difficult to compute due to the intractability of $u_t^\circ(x)$. However, the availability of a conditional vector field $u_t^\circ(x|z)$ allows for optimizing an alternative objective $\mathcal{L}_{CFM}(\theta) = \mathbb{E}_{t,q(z),p_t(x|z)}\|v_t^\theta(x) - u_t^\circ(x|z)\|_2^2$, and it has been proven that $\nabla_\theta \mathcal{L}_{FM}(\theta) = \nabla_\theta \mathcal{L}_{CFM}(\theta)$ (Lipman et al., 2022; Tong et al., 2023).

### 3.2 PROBLEM DEFINITION AND BASIC PATH PREPARATION

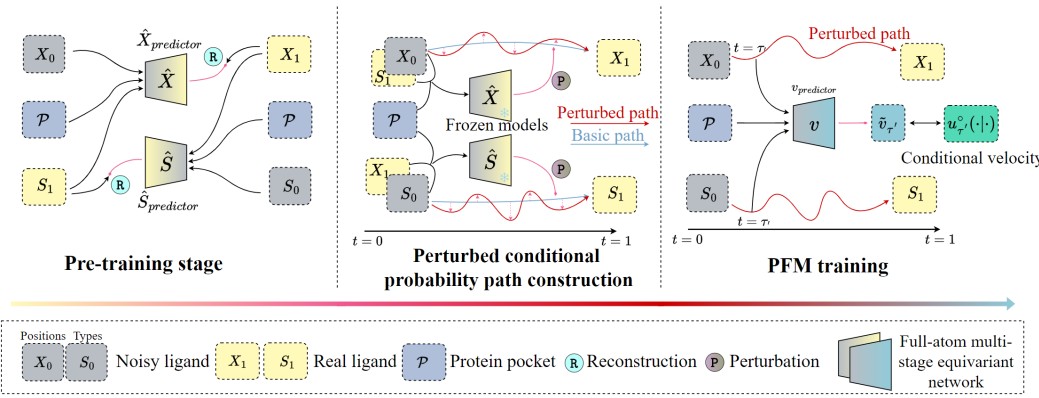

Figure 1: Specific training procedure of PFM. The **middle** is the crucial construction of the perturbed conditional probability path.

**Problem Definition** Using generative models for SBDD involves generating ligand molecules conditioned on a given protein binding site. The protein and ligand molecule can be represented as $\mathcal{P} = \{(x_i^{\mathcal{P}}, a_i^{\mathcal{P}})\}_{i=1}^{N_P}$ and $\mathcal{M} = \{(x_i^{\mathcal{M}}, a_i^{\mathcal{M}})\}_{i=1}^{N_M}$, where $N_P$ and $N_M$ refer to the number of atoms in the protein and ligand molecule, respectively. $x_i \in \mathbb{R}^3$ and $a_i \in \{1, 2, \ldots, D\}$ represent coordinate and types of the atom, respectively. For simplicity, we denote molecules as $\mathcal{M} = \{(x_i^{\mathcal{M}}, s_i^{\mathcal{M}})\}_{i=1}^{N_M}$, where $s_i \in \mathbb{R}^D$ is the continuous representation of $a_i$ and $D$ represents the number of atom types in the ligand molecule. Furthermore, since types and coordinates of atoms in the protein remain constant during the process, we only simplify the representation of the molecule as $\mathbf{M} = [X, S]$, where $X \in \mathbb{R}^{N_M \times 3}$ and $S \in \mathbb{R}^{N_M \times D}$. The task is then framed as modeling the conditional distribution $q(\mathbf{M}|\mathcal{P})$, with the number of atoms in the ligand molecule sampled from the prior distribution (Guan et al., 2023a).

**Continuous Processing of Atom Types** To map discrete type $a_i$ to a continuous space, we employ a soft one-hot encoding operation $\texttt{logit}(a_i) = s_i \in \mathbb{R}^D$, with a constant value $K > 0$, and the $j$-th value of $s_i$ is represented as:

$$s_i[j] = \begin{cases} K, & j = a_i \\ -K, & \text{otherwise,} \end{cases} \tag{1}$$

where $s_i$ can be interpreted as the logit of probabilities (Han et al., 2023).

**Basic Conditional Probability Path Preparation** Before performing path perturbation, we select the basic paths for types and coordinates of atoms. Here, for simplicity, we adopt two similar paths for $S$ and $X$:

$$p_t(S|S_0, S_1, \mathcal{P}) = \mathcal{N}(S|tS_1 + (1 - t)S_0, \sigma^2 I), \tag{2}$$

$$p_t(X|X_0, X_1, \mathcal{P}) = \mathcal{N}(X|tX_1 + (1 - t)X_0, \sigma^2 I), \tag{3}$$

where $q(S_0) = \mathcal{N}(0, K^2 I)$, $q(X_0) = \mathcal{N}(0, I)$ and subscripts on $S$ and $X$ denote time. Further details on atom type continuousization and sampling can be found in Appendix A.3.

### 3.3 Perturbed Conditional Probability Path Design

Different choices of conditional probability paths $p_t(x|z)$ and conditioning distributions $q(z)$ can lead to distinct optimization results (Tong et al., 2023; Song et al., 2024; Klein et al., 2024). Here, we introduce a novel path that does not rely on specific optimization objectives. Specifically, we propose perturbing simple conditional probability paths by incorporating pocket binding site information and atom type-coordinate coupled information to construct a new conditional probability path, which we term a "perturbed conditional probability path". The form is as follows:

$$p_t(\mathbf{M}|\mathbf{M}_0, \mathbf{M}_1, \mathcal{P}) = \mathcal{N}(\mathbf{M}|t\mathbf{M}_1 + (1-t)\mathbf{M}_0 + t(1-t)W \odot \hat{\mathbf{M}}(\mathbf{M}_1, \mathbf{M}_0, \mathcal{P}), \sigma^2 I), \quad (4)$$

$$q(\mathbf{M}_0, \mathbf{M}_1|\mathcal{P}) = q(\mathbf{M}_0)q(\mathbf{M}_1|\mathcal{P}), \quad (5)$$

where $W \in \mathbb{R}^{N_M \times (3+D)}$ represents the weight matrix. $\hat{\mathbf{M}}(\cdot)$ is a function, which, together with $W$, is required to ensure that $p_t(\mathbf{M}|\mathbf{M}_0, \mathbf{M}_1, \mathcal{P})$ can be integrated over $\mathbf{M}_0$ and $\mathbf{M}_1$ to obtain the marginal probability $p_t(\mathbf{M}|\mathcal{P})$. $\odot$ denotes the Hadamard product. In the independent coupling represented by Eq.5, translational invariance in coordinate $X$, where $p(X+\mathbf{t}) = p(X)$ and $\mathbf{t}$ denotes the translation, can lead to non-normalizable probability distributions. To address this issue, we use the centroid of the pocket binding site as the origin (Garcia Satorras et al., 2021). In practice, we further employ the following simplified perturbed conditional probability path, as illustrated in Fig.1:

$$p_t(\mathbf{M}|\mathbf{M}_0, \mathbf{M}_1, \mathcal{P}) = \mathcal{N}([X, S]|[tX_1 + (1-t)X_0 + \xi t(1-t)\hat{X}(S_1, X_0, \mathcal{P}),$$
$$tS_1 + (1-t)S_0 + \xi t(1-t)\hat{S}(X_1, S_0, \mathcal{P})], \sigma^2 I), \quad (6)$$

where $\hat{X}(\cdot)$ and $\hat{S}(\cdot)$ are functions, with $\hat{X}$ and $\hat{S}$ in the following referring to the perturbations from computation. In practice, we find that using neural networks to generate these perturbations works well. The magnitude of the perturbation is jointly controlled by $\xi$ and $t(1-t)$.

Eq.4 and 6 describe the conditional flow of each element in the matrix $\mathbf{M}$ or $[X, S]$. The details of $\hat{X}$ and $\hat{S}$ involved are introduced at the end of Sec.3.4. According to Sec.3.1 and **Theorem 3** of Lipman et al. (2022), the conditional vector field can be expressed as follows:

$$u_t^\circ(\mathbf{M}|\mathbf{M}_0, \mathbf{M}_1, \mathcal{P}) = [X_1 - X_0 + \xi(1-2t)\hat{X}, S_1 - S_0 + \xi(1-2t)\hat{S}]. \quad (7)$$

Notably, $p_t(\mathbf{M}|\mathbf{M}_0, \mathbf{M}_1, \mathcal{P})$ can be efficiently sampled, and $u_t^\circ(\mathbf{M}|\mathbf{M}_0, \mathbf{M}_1, \mathcal{P})$ can be easily computed, which makes the optimization of $\theta$ on $\mathcal{L}_{CFM}$ feasible. As $\sigma \to 0$, the marginal boundary probabilities converge to $q(\mathbf{M}_0)$ and $q(\mathbf{M}_1|\mathcal{P})$ according to **Proposition 3.3** in Tong et al. (2023).

Further details on constructing flows for the SBDD task can be found in Appendix A.

### 3.4 Training Objective

**Flow Matching Loss** According to Sec. 3.1, the velocity-field loss can be expressed as follows:

$$\mathcal{L}_M = \mathbb{E}_{t, q(\mathbf{M}_0, \mathbf{M}_1|\mathcal{P}), p_t(\mathbf{M}|\mathbf{M}_0, \mathbf{M}_1, \mathcal{P})} \|v_t^\theta(\mathbf{M}|\mathcal{P}) - u_t^\circ(\mathbf{M}|\mathbf{M}_0, \mathbf{M}_1, \mathcal{P})\|_F^2, \quad (8)$$

where $t \sim \mathcal{U}(0, 1)$. The detailed reparameterization and analysis are provided in Appendix A.1. However, $\mathcal{L}_M$ focuses solely on individual atoms, lacking consideration for local molecular structures and interactions with the protein. Here, we propose incorporating two simple auxiliary losses to account for these considerations.

**Local Loss** From an intramolecular perspective, it is important to incorporate insights into local structure, such as bond lengths, bond angles, and torsion angles. For simplicity, we introduce only the bond length loss:

$$\mathcal{L}_B = \mathbb{E}_{t, q(\mathbf{M}_0, \mathbf{M}_1|\mathcal{P}), p_t(\mathbf{M}|\mathbf{M}_0, \mathbf{M}_1, \mathcal{P})} \sum_{i=1}^{N_B} \|\tilde{l_B} - l_B\|, \quad (9)$$

where $N_B$ denotes the number of bonds in the ligand molecule, and $l_B$ is ground truth of bond length.

**Clash Loss** From the perspective of molecule–pocket interactions, steric clashes are a critical consideration. Therefore, we introduce a clash loss to penalize atoms that intrude into the protein interior. Following Guan et al. (2023b), we choose $\{x \in \mathbb{R}^3 : \mathcal{S}(x) = \gamma\}$ where $\mathcal{S}(x) = -\tilde{\sigma} \ln \left( \sum_{j=1}^{N_P} \exp \left( -\|x - x_j^{\mathcal{P}}\|^2 / \tilde{\sigma} \right) \right)$ as the descriptor of the protein surface. Subsequently, we define a clash loss as:

$$\mathcal{L}_C = \mathbb{E}_{t, q(\mathbf{M}_0, \mathbf{M}_1 | \mathcal{P}), p_t(\mathbf{M} | \mathbf{M}_0, \mathbf{M}_1, \mathcal{P})} \sum_{i=1}^{N_M} \max \left( 0, \gamma - \mathcal{S}(\tilde{x}_i^{\mathcal{M}}) \right). \tag{10}$$

The overall loss function for PFM is thus expressed as:

$$\mathcal{L}_{PFM} = \mathcal{L}_M + \mu \mathcal{L}_B + \nu \mathcal{L}_C. \tag{11}$$

**Perturbation Network and Reconstruction Loss** To enhance the stability of PFM training and reduce memory usage, we leverage pretrained predictors for the perturbations $\hat{X}$ and $\hat{S}$ as shown in Fig.1. Specifically, we introduce a molecular reconstruction task given protein pocket $\mathcal{P}$ and atom types $S$ or coordinates $X$ of ligand. Their corresponding loss functions are defined as:

$$\mathcal{L}_{\hat{X} predictor} = \mathbb{E}_{q(\mathbf{M} | \mathcal{P}), q(\mathbf{M}_0)} \|\hat{X}(S, X_0, \mathcal{P}) - X\|_F^2, \tag{12}$$

$$\mathcal{L}_{\hat{S} predictor} = \mathbb{E}_{q(\mathbf{M} | \mathcal{P}), q(\mathbf{M}_0)} \|\hat{S}(X, S_0, \mathcal{P}) - S\|_F^2, \tag{13}$$

where $\mathbf{M} = [X, S]$. The pretrained predictors introduce meaningful and reasonable perturbations into the basic probability paths. First, pretraining aligns their outputs with the numerical scale of the basic path. Second, as intended, the perturbations inject information from the pocket and atom type or coordinate. Most importantly, they provide a coarse estimate of the molecule that biases the conditional path, which intuitively yields more plausible and more discriminative marginal paths. Extensive experiments validate the effectiveness of this design. However, this training strategy raises a concern regarding the difference between paths with mean $[tX_1 + (1-t)X_0 + \xi t(1-t)\hat{X}]$ and $[tX_1 + (1-t)X_0 + \xi t(1-t)X_1]$. We address it in Sec.4.3.

## 3.5 FULL-ATOM MULTI-STAGE EQUIVARIANT NETWORK

Inspired by recent advancements in equivariant neural networks (Satorras et al., 2021; Liu et al., 2022b; Liao et al., 2023) and protein-ligand binding affinity prediction (Jiang et al., 2021), a multi-stage approach is proposed to update features and positions.

First, three atom-level $k$-nearest neighbor (knn) graphs are constructed: $\mathcal{G}_{\mathcal{P}}$, $\mathcal{G}_{\mathcal{M}}$, and $\mathcal{G}_{\mathcal{I}}$ representing the protein, ligand molecule, and protein-ligand interaction graphs, respectively. The value of $k$ differs for each graph. After embedding the initial node features into the same dimension for both protein and molecule atoms, the protein atom features are updated as follows in each layer:

$$m_{ij} \leftarrow \phi_d \left( \|x_i - x_j\|, e_{ij} \right), \tag{14}$$

$$h_{i,1}^{\mathcal{P}} \leftarrow h_i^{\mathcal{P}} + \sum_{j \in \mathcal{N}_{\mathcal{P}}(i)} \phi_{m_{\mathcal{P}}} \left( h_i^{\mathcal{P}}, h_j^{\mathcal{P}}, m_{ij} \right), \tag{15}$$

where $h_{i,1}^{\mathcal{P}}$ represents the updated feature of node $i$ in graph $\mathcal{G}_{\mathcal{P}}$, $\mathcal{N}_{\mathcal{P}}(i)$ denotes the set of neighboring nodes' indices of node $i$ in $\mathcal{G}_{\mathcal{P}}$, $e_{ij}$ represents edge $ij$ in related $\mathcal{G}$, and $m_{ij}$ denotes message. Since the network ultimately obtains atom representations in ligand molecules, the updating strategy in $\mathcal{G}_{\mathcal{M}}$ differs slightly from that in $\mathcal{G}_{\mathcal{P}}$. **In the first stage**, atomic interactions within the ligand molecule are considered, and the type features and positions are updated as follows:

$$h_{i,1}^{\mathcal{M}} \leftarrow h_i^{\mathcal{M}} + \sum_{j \in \mathcal{N}_{\mathcal{M}}(i)} \phi_{m_{\mathcal{M}}} \left( h_i^{\mathcal{M}}, h_j^{\mathcal{M}}, m_{ij}, t \right), \tag{16}$$

$$\Delta x_{i,1}^{\mathcal{M}} \leftarrow \sum_{j \in \mathcal{N}_{\mathcal{M}}(i)} (x_j^{\mathcal{M}} - x_i^{\mathcal{M}}) \phi_{x_{\mathcal{M}}} \left( h_{i,1}^{\mathcal{M}}, h_{j,1}^{\mathcal{M}}, d_{ij}, t \right), \tag{17}$$

$$x_{i,1}^{\mathcal{M}} \leftarrow x_i^{\mathcal{M}} + \Delta x_{i,1}^{\mathcal{M}}, \tag{18}$$

where $d_{ij}$ is the Euclidean distance between nodes $i$ and $j$.

**In the second stage**, atomic interactions between the ligand molecule and the protein are considered, and the type features and positions are updated as follows:

$$h_{i,2}^{\mathcal{I}} \leftarrow h_{i,1}^{\mathcal{I}} + \sum_{j \in \mathcal{N}_{\mathcal{I}}(i)} \phi_{m_{\mathcal{I}}} \left( h_{i,1}^{\mathcal{I}}, h_{j,1}^{\mathcal{I}}, m_{ij}, t \right), \tag{19}$$

$$\Delta x_{i,2}^{\mathcal{M}} \leftarrow \sum_{j \in \mathcal{N}_{\mathcal{I}}(i)} (x_j^{\mathcal{P}} - x_{i,1}^{\mathcal{M}}) \phi_{x_{\mathcal{I}}} \left( h_{i,2}^{\mathcal{M}}, h_{j,2}^{\mathcal{P}}, d_{ij}, t \right), \tag{20}$$

$$x_{i,2}^{\mathcal{M}} \leftarrow x_{i,1}^{\mathcal{M}} + \Delta x_{i,2}^{\mathcal{M}}. \tag{21}$$

Here, Eq.20 distinguishes the atomic origin. During both stages, the subscripts on $h$ and $x$ (e.g., 1, 2) denote updates within the same layer. $m_{ij}$ and $d_{ij}$ change dynamically due to updating.

After $L$ layers of updates, the network outputs the prediction, which are then passed to the loss function $\mathcal{L}_{PFM}$ for parameter optimization. The same network structure is adopted for pretrained predictors. Further details on predictors pretraining and PFM training can be found in Appendix C, while procedures for training and sampling are provided in Algorithm 1 and Algorithm 2.

### 3.6 GUIDED GENERATION

To further improve molecular generation quality, inspired by previous work (Dhariwal & Nichol, 2021; Song et al., 2020), we employ two generic approaches for binding affinity-guided generation: **classifier guidance** and **classifier-free guidance** (Ho & Salimans, 2022).

For classifier guidance, we add an extra output head to the network for binding affinity prediction, with $\mathcal{L}_{vs} = \mathbb{E}_{t,p_t(\mathbf{M}|\mathcal{P})}\|\hat{y} - y\|^2$ and $\hat{y} = (\sum_{i=1}^{N_M} \texttt{sigmoid}(\texttt{MLP}(h_{i,2L}^{\mathcal{M}})))/N_M$, where $y$ denotes the binding affinity label (AutoDock Vina scores on the CrossDocked2020 dataset scaled to $[0, 1]$) and $\hat{y}$ is the prediction. Following Lipman et al. (2024), guidance is applied as $\tilde{v}_t^{\theta,\phi}(\mathbf{M}|\mathcal{P}, y) = v_t^\theta(\mathbf{M}|\mathcal{P}) + w\nabla \log p_{Y|t}^\phi(y|\mathbf{M}, \mathcal{P})$, where $w$ is the product of a time-dependent coefficient and a hyperparameter. For classifier-free guidance, we add an extra input to the network: an embedding vector of the condition $y$. Formally, we use $\tilde{v}_t^\theta(\mathbf{M}|\mathcal{P}, y) = (1-\tilde{w})v_t^\theta(\mathbf{M}|\mathcal{P}, \emptyset) + \tilde{w}v_t^\theta(\mathbf{M}|\mathcal{P}, y)$. Despite the presence of estimation error, experiments demonstrate the effectiveness of these guidance strategies. Detailed analysis is provided in Appendix A.4.

## 4 EXPERIMENTS

### 4.1 EXPERIMENTAL SETUP

**Dataset** Following the previous work (Luo et al., 2021; Peng et al., 2022), we train and evaluate PFM on CrossDocked2020 dataset (Francoeur et al., 2020). The data preprocessing and splitting methods used by Guan et al. (2023a) and Luo et al. (2021) are adopted, refining the 22.5 million docked binding complexes to retain high-quality docking poses (RMSD between the docked pose and the ground truth $< 1\,\text{Å}$) and diverse proteins (sequence identity 30%). Ultimately, 99,900 qualified complexes are utilized for training, and 100 novel proteins are selected for testing.

**Baseline** In this study, the proposed method is compared with several recent representative approaches: **LiGAN** (Ragoza et al., 2022) is a CVAE model implemented on atomic density grids. **AR** (Luo et al., 2021), **Pocket2Mol** (Peng et al., 2022) and **GraphBP** (Liu et al., 2022a) generate molecules autoregressively by leveraging contextual information. **TargetDiff** (Guan et al., 2023a), **DecomposeDiff** (Guan et al., 2023b), and **IPDiff** (Huang et al., 2024) are diffusion-based models designed for molecular generation.

**Evaluation** The generated molecules are evaluated from two perspectives: **Target Binding Results** and **Molecular Conformations and Properties**.

· **Target Binding Results** The mean and median of *Vina Score*, *Vina Min*, *Vina Dock*, and *High Affinity* are used to evaluate binding affinities (Luo et al., 2021; Ragoza et al., 2022). *Vina Score*

directly estimates the binding affinity of generated molecules to the target, *Vina Min* performs local minimization before estimation, *Vina Dock* involves a re-docking process reflecting the best possible binding affinity, and *High Affinity* indicates the percentage of molecules generated for each test protein that bind better than the reference molecules. In addition, *Clash Ratio* and *RMSD* are considered to evaluate the geometry of docking poses. Specifically, *CCA.* denotes the ratio of atoms that clash with protein atoms to the total number of atoms, and *CM.* indicates the ratio of molecules with clashes (Lin et al., 2024). $\% < 2\text{Å}$ denotes the percentage of generated molecules with RMSD less than 2Å from the redocked poses, serving as a measure of the model's ability to learn the docking poses (McNutt et al., 2021; Qu et al., 2024).

· **Molecular Conformation and Properties** Molecular conformation is evaluated by measuring the Jensen-Shannon divergences (JSD) between the bond distance distributions of reference and generated molecules. For molecular properties, *QED*, *SA*, *Diversity*, *Similarity*, *Validity*, and *Uniqueness* are used (Luo et al., 2021; Ragoza et al., 2022; O Pinheiro et al., 2023; Reidenbach, 2024). *QED* quantifies the drug-likeness of molecules, *SA* evaluates their synthetic accessibility, and *Diversity* represents the average pairwise dissimilarity between the molecules generated for a given pocket binding site. *Similarity* measures the average similarity between generated molecules and their references. *Validity* refers to the proportion of generated molecules that can be reconstructed by RDKit and have binding affinities $< 0$. *Uniqueness* indicates the proportion of valid molecules with different canonical SMILES.

## 4.2 MAIN RESULTS

**Target Binding Results** As shown in Tab.1, for **binding affinity**, PFM significantly outperforms the autoregressive model Pocket2Mol, with notable improvements of 38.5%, 18.1%, and 16.4% in the average Vina Score, Vina Min, and Vina Dock, respectively. Compared with IPDiff, PFM achieves competitive results without introducing additional prior knowledge (e.g., binding affinity prediction tasks), showing a 10.9% improvement in average Vina Score, a slight increase of 1.7% in average Vina Min, and a minor decrease of 2.9% in average Vina Dock. It is important to note that, as described in Sec.4.1, Vina Score best reflects the model's molecular generation and docking ability. In addition, larger generated molecules introduced by the model could potentially lead to better Vina Scores. However, as shown by the average number of atoms in Tab.1, PFM does not appear to benefit significantly from it. For **geometry evaluation**, PFM outperforms other diffusion-based methods in Clash Ratio and RMSD. LiGAN achieves the best clash ratio, primarily because its grid-based modeling effectively avoids most steric clashes. With guidance enabled, the model's performance improves overall, as expected. $\text{PFM}_{cfg}$ outperforms $\text{PFM}_{cg}$, possibly due to the conditional–unconditional differential guidance within a single network, which intuitively reduces gradient noise and makes scale calibration easier.

Table 1: Summary of target binding results of reference molecules and molecules generated by our model and other non-diffusion (**Others**) and diffusion-based (**Diffusion**) baselines. (↑) / (↓) denotes a larger / smaller number is better. Top 3 results are highlighted with **bold text** and underlined text, respectively.

| Methods | | Vina Score (↓) | | Vina Min (↓) | | Vina Dock (↓) | | High Affinity (↑) | | Clash Ratio (↓) | | RMSD (↑) | # Atoms /mol. |
| --- | --- | --- | --- | --- | --- | --- | --- | --- | --- | --- | --- | --- | --- |
| | | Avg. | Med. | Avg. | Med. | Avg. | Med. | Avg. | Med. | CCA. | CM. | % < 2Å | Avg. |
| Reference | | -6.36 | -6.46 | -6.71 | -6.49 | -7.45 | -7.26 | - | - | - | - | 34.0 | 22.8 |
| **Others** | liGAN | - | - | - | - | -6.33 | -6.20 | 21.1% | 11.1% | **0.0096** | **0.0718** | 43.8 | 19.7 |
| | GraphBP | - | - | - | - | -4.80 | -4.70 | 14.2% | 6.7% | 0.8634 | 0.9974 | - | 23.1 |
| | AR | -5.75 | -5.64 | -6.18 | -5.88 | -6.75 | -6.62 | 37.9% | 31.0% | - | - | - | - |
| | Pocket2Mol | -5.14 | -4.70 | -6.42 | -5.82 | -7.15 | -6.79 | 48.4% | 51.0% | 0.0576 | 0.4499 | 30.8 | 17.7 |
| **Diffusion** | TargetDiff | -5.47 | -6.30 | -6.64 | -6.83 | -7.80 | -7.91 | 58.1% | 59.1% | 0.0483 | 0.4920 | 29.4 | 24.2 |
| | DecompDiff | -5.67 | -6.04 | -7.04 | -7.09 | -8.39 | -8.43 | 64.4% | 71.0% | 0.0462 | 0.5248 | 23.9 | 20.9 |
| | IPDiff | -6.42 | -7.01 | -7.45 | -7.48 | -8.57 | -8.51 | 69.5% | 75.5% | 0.0313 | 0.3463 | - | 24.5 |
| **FM** | PFM | -7.12 | -7.18 | -7.58 | -7.32 | -8.32 | -8.26 | 72.1% | 76.5% | 0.0145 | 0.2040 | 36.7 | 22.6 |
| | $\text{PFM}_{cg}$ | 7.06 | 7.41 | -7.78 | -7.57 | -8.42 | -8.43 | 68.4% | 72.5% | 0.0105 | 0.1675 | 39.3 | 22.7 |
| | $\text{PFM}_{cfg}$ | **-7.83** | **-7.88** | **-8.21** | **-8.00** | **-8.64** | **-8.63** | **74.7%** | **84.5%** | 0.0108 | 0.1633 | **53.0** | 22.7 |

**Molecular Conformation and Properties** JSD values between bond distance distributions of reference and generated molecules are calculated and summarized in Tab.2. The results demonstrate that molecules generated by PFM achieved competitive results compared with IPDIFF. Regarding **molecular properties**, PFM achieves superior or competitive performance across most metrics

compared with diffusion-based methods, and these results are not markedly affected under binding-affinity guidance. Although PFM shows depressed SA performance, as shown in the ablation studies (Tabs.4 and 5), this primarily reflects a QED–SA trade-off induced by the perturbed conditional probability path and the $\mathcal{L}_B$ term, rather than a direct degradation. Nonetheless, according to Ertl & Schuffenhauer (2009), the SA scores of molecules generated by PFM remain within an acceptable range and do not hinder the rough screening of drug discovery (Guan et al., 2023a;b).

Table 2: Jensen-Shannon divergence between bond distance distributions of the reference molecules and the generated molecules, and lower values indicate better performances. "-", "=", and ":" represent single, double, and aromatic bonds, respectively. Top 3 results are highlighted with **bold text** and underlined text, respectively.

| Bond | liGAN | GraphBP | AR | Pocket2 Mol | Target Diff | Decomp Diff | IP Diff | PFM |
|------|-------|---------|-----|-------------|-------------|-------------|---------|-----|
| C−C | 0.601 | 0.368 | 0.609 | 0.496 | 0.369 | **0.359** | 0.386 | 0.395 |
| C=C | 0.665 | 0.530 | 0.620 | 0.561 | 0.505 | 0.537 | 0.245 | **0.172** |
| C−N | 0.634 | 0.456 | 0.474 | 0.416 | 0.363 | 0.344 | 0.298 | **0.266** |
| C=N | 0.749 | 0.693 | 0.635 | 0.629 | 0.550 | 0.584 | 0.238 | **0.147** |
| C−O | 0.656 | 0.467 | 0.492 | 0.454 | 0.421 | 0.376 | 0.366 | **0.317** |
| C=O | 0.661 | 0.471 | 0.558 | 0.516 | 0.461 | 0.374 | **0.353** | 0.411 |
| C:C | 0.497 | 0.407 | 0.451 | 0.416 | 0.263 | 0.251 | **0.169** | 0.188 |
| C:N | 0.638 | 0.689 | 0.552 | 0.487 | 0.235 | 0.269 | **0.128** | 0.187 |

Table 3: Summary of molecular conformation and properties of reference molecules and molecules generated by our model and other baselines. (↑) / (↓) denotes a larger / smaller number is better. Top 3 results are highlighted with **bold text** and underlined text, respectively.

| Methods | QED (↑) | | SA (↑) | | Diversity (↑) | | Similarity (↓) | Validity (↑) | Uniqueness (↑) |
|---------|---------|------|--------|------|---------------|------|----------------|--------------|----------------|
| | Avg. | Med. | Avg. | Med. | Avg. | Med. | Avg. | Avg. | Avg. |
| Reference | 0.48 | 0.47 | 0.73 | 0.74 | - | - | - | 100% | - |
| liGAN | 0.39 | 0.39 | 0.59 | 0.57 | 0.66 | 0.67 | 0.22 | - | 87.28% |
| GraphBP | 0.43 | 0.45 | 0.49 | 0.48 | **0.79** | **0.78** | **0.15** | - | **100%** |
| Pocket2Mol | **0.56** | **0.57** | **0.74** | **0.75** | 0.69 | 0.71 | 0.26 | **98.31%** | **100%** |
| TargetDiff | 0.48 | 0.48 | 0.58 | 0.58 | 0.72 | 0.71 | 0.30 | 90.35% | 99.63% |
| DecompDiff | 0.45 | 0.43 | 0.61 | 0.60 | 0.68 | 0.68 | 0.34 | 71.96% | 99.99% |
| IPDiff | 0.52 | 0.53 | 0.61 | 0.59 | 0.74 | 0.73 | 0.18 | 88.16% | **100%** |
| PFM | 0.49 | 0.50 | 0.51 | 0.51 | 0.73 | 0.71 | 0.20 | 95.12% | **100%** |
| PFM$_{cg}$ | 0.49 | 0.50 | 0.51 | 0.52 | 0.73 | 0.72 | 0.19 | 95.32% | **100%** |
| PFM$_{cfg}$ | 0.47 | 0.49 | 0.53 | 0.52 | 0.73 | 0.73 | 0.20 | 94.40% | **100%** |

**Faster Sampling and Performance Gains from Perturbation** As shown in Tab.7 in appendix, PFM achieves a 21.3x speed-up in sampling, benefiting from the ODE samplers and the network. Specifically, the multi-stage design enables more flexible graph structures and message-passing for both intra- and intermolecular graphs-e.g., reducing the number of edges in the protein graph can substantially cut the majority of message-passing without affecting overall performance. The proposed perturbed conditional probability path markedly improves docking energy and introduces a QED–SA trade-off (as indicated by BFM* and PFM*), while a simple auxiliary loss further enhances effectiveness. Additional potential for faster sampling is discussed in Appendix D.4.

## 4.3 ABLATION STUDIES

PFM introduces an FM-based molecular generation approach for SBDD, featuring a novel perturbed conditional probability path and a well-designed loss function. A comprehensive ablation validates the effectiveness of these components. Further results and analyses on (i) reducing the number of sampling steps, (ii) differences in predicted fields induced by the perturbed conditional probability paths, (iii) selection of the basic path, and (iv) the impact of perturbation coefficients are provided in Appendix D.4 and D.2.

Table 4: The influence of perturbed conditional probability path.

| Methods | Vina Score (↓) | | Vina Min (↓) | | QED (↑) | | SA (↑) | |
| | Avg. | Med. | Avg. | Med. | Avg. | Med. | Avg. | Med. |
|---|---|---|---|---|---|---|---|---|
| BFM* | -4.27 | -4.94 | -6.04 | -5.94 | 0.46 | 0.47 | 0.50 | 0.51 |
| BFM | -5.77 | -6.27 | -6.76 | -6.71 | 0.47 | 0.48 | 0.54 | 0.54 |
| BFM-Ref | -6.48 | -6.89 | -7.27 | -7.10 | **0.49** | **0.50** | 0.51 | 0.51 |
| LPFM | -6.72 | -6.88 | -7.37 | -7.13 | **0.49** | **0.50** | 0.51 | 0.50 |
| RPFM | -6.86 | -7.17 | -7.51 | **-7.33** | 0.48 | 0.49 | 0.52 | 0.52 |
| PFM* | -6.48 | -6.49 | -6.79 | -6.45 | 0.43 | 0.43 | **0.57** | **0.57** |
| PFM | **-7.12** | **-7.18** | **-7.58** | -7.32 | **0.49** | **0.50** | 0.51 | 0.51 |

**Effect of Perturbed Conditional Probability Path**  The perturbed conditional probability path in PFM is hypothesized to improve molecular generation. Tab.4 evaluates path choices that share Gaussian variance and differ only in the mean (for the path on $X$): (i) **BFM**: $[tX_1 + (1-t)X_0]$, (ii) **BFM-Ref**: $[tX_1 + (1-t)X_0 + \xi t(1-t)X_1]$, (iii) **LPFM**: $[tX_1 + (1-t)X + \zeta t(1-t)^2\hat{X}]$ and (iv) **RPFM**: $[tX_1 + (1-t)X + \zeta t^2(1-t)\hat{X}]$. Note that **BFM\*** and **PFM\*** indicate models trained with consideration of $\mathcal{L}_M$ only. The results demonstrate that the proposed perturbed paths improve overall molecular generation performance (as shown by comparisons of PFM with BFM and BFM-Ref, or PFM\* with BFM\*) and confirm the effectiveness of PFM under different perturbation strategies (as evidenced by comparisons of LPFM, RPFM, PFM with BFM and BFM-Ref).

**Impact of the Loss Function**  The proposed loss function significantly enhances PFM's molecular generation performance. Tab.5 presents an ablation study of the loss terms, indicating that $\mathcal{L}_B$ contributes a lot to performance improvement. $\mathcal{L}_C$ provides slight benefits to molecular generation performance only when combined with $\mathcal{L}_B$ and $\mathcal{L}_M$.

Table 5: The effect of different loss terms.

| $\mathcal{L}_M$ | $\mathcal{L}_B$ | $\mathcal{L}_C$ | Vina Score (↓) | | Vina Min (↓) | | QED (↑) | | SA (↑) | |
| | | | Avg. | Med. | Avg. | Med. | Avg. | Med. | Avg. | Med. |
|---|---|---|---|---|---|---|---|---|---|---|
| ✔ | ✗ | ✗ | -6.48 | -6.49 | -6.79 | -6.45 | 0.43 | 0.43 | **0.57** | **0.57** |
| ✔ | ✔ | ✗ | -6.88 | -7.06 | -7.43 | -7.18 | 0.49 | 0.50 | 0.52 | 0.51 |
| ✔ | ✗ | ✔ | -6.45 | -6.48 | -6.83 | -6.51 | 0.42 | 0.42 | **0.57** | 0.56 |
| ✔ | ✔ | ✔ | **-7.12** | **-7.18** | **-7.58** | **-7.32** | 0.49 | 0.50 | 0.51 | 0.51 |

## 4.4 MORE EXPERIMENTS ON MOLECULAR DATASETS

We further validate the effectiveness of the perturbed conditional probability path design. Specifically, for the SBDD task, we conduct preliminary evaluation on BindingMOAD dataset (Hu et al., 2005). In addition, for molecular generation, experiments on QM9 dataset (Ruddigkeit et al., 2012) further support the method's effectiveness. See Appendix D.3 for detailed results.

## 5 CONCLUSIONS

In this paper, a method called PFM for the SBDD task is proposed for the first time. Building on a basic FM framework, PFM enhances molecular generation performance by designing a more differentiated perturbed path, which incorporates pocket binding site information and atom type-coordinate coupled information into a basic conditional probability path. Additionally, the introduction of a flexible full-atom multi-stage equivariant network and a simple auxiliary loss also contribute to improved results. Experiments demonstrate the efficiency of the model and the effectiveness of its components, and highlight PFM's potential to generate competitive molecules with faster sampling.

ETHICS STATEMENT

This work adheres to the ICLR Code of Ethics.

REPRODUCIBILITY STATEMENT

All datasets used are public; code and processing details are provided in the submission.

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

## A   FLOW MATCHING IN THE SBDD TASK

### A.1   REPARAMETERIZATION OF $\mathcal{L}_M$

Within the flow matching framework, data-prediction, noise-prediction, and velocity-prediction can each be trained with neural networks to predict the marginal velocity field. After adding perturbations, the most straightforward choice is to use velocity-prediction as in Eq.8; however, preliminary experiments indicate that this approach struggles to produce effective molecules. We therefore adopt data-prediction. Let the network's predictions for $X_1$ and $S_1$ be $[\tilde{X}, \tilde{S}]$, and Substitute Eq.7 into Eq.8:

$$
\begin{aligned}
\mathcal{L}_M(\theta) &= \mathbb{E}_{t, q(\mathbf{M}_0, \mathbf{M}_1 | \mathcal{P}), p_t(\mathbf{M} | \mathbf{M}_0, \mathbf{M}_1, \mathcal{P})} \| [v_t^{(\theta, X)}(\mathbf{M} | \mathcal{P}), v_t^{(\theta, S)}(\mathbf{M} | \mathcal{P})] \\
&\quad - [X_1 - X_0 + \xi(1 - 2t)\hat{X}, S_1 - S_0 + \xi(1 - 2t)\hat{S}] \|_F^2 \\
&= \mathbb{E}_{t, q(\mathbf{M}_0, \mathbf{M}_1 | \mathcal{P}), p_t(\mathbf{M} | \mathbf{M}_0, \mathbf{M}_1, \mathcal{P})} \| [\tilde{X}_1^{(\theta)}(t, \mathbf{M}, \mathcal{P}) - X_0 + \xi(1 - 2t)\hat{X}, \\
&\quad \tilde{S}_1^{(\theta)}(t, \mathbf{M}, \mathcal{P}) - S_0 + \xi(1 - 2t)\hat{S}] - [X_1 - X_0 + \xi(1 - 2t)\hat{X}, S_1 - S_0 + \xi(1 - 2t)\hat{S}] \|_F^2 \\
&= \mathbb{E}_{t, q(\mathbf{M}_0, \mathbf{M}_1 | \mathcal{P}), p_t(\mathbf{M} | \mathbf{M}_0, \mathbf{M}_1, \mathcal{P})} \| [\tilde{X}_1^{(\theta)}(t, \mathbf{M}, \mathcal{P}) - X_1, \tilde{S}_1^{(\theta)}(t, \mathbf{M}, \mathcal{P}) - S_0] \|_F^2 \\
&= \mathbb{E}_{t, q(\mathbf{M}_0, \mathbf{M}_1 | \mathcal{P}), p_t(\mathbf{M} | \mathbf{M}_0, \mathbf{M}_1, \mathcal{P})} \| [\tilde{X} - X_1, \tilde{S} - S_1] \|_F^2.
\end{aligned}
$$

It is important to note that, taking the vector field of position $X$ as an example, the perturbation term contributing to the predicted vector field in the second equality is actually $\hat{X}(\tilde{S}, X_0, \mathcal{P})$. Strictly enforcing this term increases training complexity and degrades the quality of the final generation. Our simple loss function provided is sufficient to optimize the prediction, and this simplification does not compromise the inference process. The results in Tab.6 validate the effectiveness of our simplification.

Moreover, during sampling, considering the case $\sigma \to 0$ in Eq.6 and neglecting the noise, the marginal velocity can be written as:

$$u_t^\circ(X|\mathcal{P}) = -\frac{1}{1-t}(X_t - \mathbb{E}[X_1|\mathbf{M_t}, \mathcal{P}]) + \xi(1-t)\mathbb{E}[\hat{X}|\mathbf{M_t}, \mathcal{P}]. \tag{22}$$

Therefore, directly using sampled values to compute the perturbation term introduces an error. In practice, this is acceptable—partly because data-prediction inherently learns a denoising map from the current noisy state to the underlying data—and it does not catastrophically degrade the quality of the generated molecules.

Table 6: The influence of loss function and conditioning distribution. PFM-True Loss (**PFM-TL**) indicates the use of $\hat{X}(\tilde{S}, X_0, \mathcal{P})$ and $\hat{S}(\tilde{X}, S_0, \mathcal{P})$ during training, while **PFM-IC** represents the use of independent coupling during the training process.

| Methods | Vina Score ($\downarrow$) | | Vina Min ($\downarrow$) | | QED ($\uparrow$) | | SA ($\uparrow$) | | Loss ($\downarrow$) | |
| | Avg. | Med. | Avg. | Med. | Avg. | Med. | Avg. | Med. | Training | Test |
|---|---|---|---|---|---|---|---|---|---|---|
| PFM-TL | -6.83 | -6.98 | -7.49 | -7.23 | 0.47 | 0.47 | 0.50 | 0.51 | - | - |
| PFM-IC | -4.42 | -4.76 | -7.58 | -7.16 | 0.49 | 0.50 | 0.42 | 0.39 | 5.48 | 9.83 |
| PFM | -7.12 | -7.18 | -7.58 | -7.32 | 0.49 | 0.50 | 0.51 | 0.51 | 1.27 | 2.28 |

## A.2 MODIFICATION OF INDEPENDENT COUPLING (EQ.5)

Tong et al. (2023) used mini-batch optimal transport to improve Flow Matching (OT-FM), while Klein et al. (2024) extended this with Equivariant Optimal Transport Flow Matching (EOT-FM) to provide improved conditioning distributions. These approaches significantly shorten the path length during training, empirically improving both training stability and generation quality. However, applying EOT to the SBDD task poses practical challenges. Specifically, variations in atom counts among ligands within a batch make mini-batch EOT infeasible. Inspired by the explanation of **Discrete Optimal Transport** in Appendix B.1 of Klein et al. (2024), we propose a method to shorten the path length and devise a corresponding sampling strategy. When the cost function $c(X_0, X_1) = d(X_0, X_1)$, where $d$ denotes the Euclidean distance, the sampling process is defined as:

$$\text{Sampling pairs: } (X_0, X_1^*) = (X_0, g^* \cdot X_1), \quad g^* := \arg\min_{g \in G} c(X_0, g \cdot X_1), \tag{23}$$

where $X_0$ and $X_1$ sampled according to Eq.5, and $G$ is a compact (topological) group acting on an $n$-dimension Euclidean space $X$ via isometries. This includes the symmetric group $S_n$, the orthogonal group $O_n$, and all their subgroups. Fig.2 provides a more intuitive depiction of the differences in basic conditional probability paths (flow matching differences) during training. In practice, for simplicity, only the symmetric group $S_{N_M}$ is considered, and $s^*$ is obtained using Hungarian algorithm (Kuhn, 1955). The impact of the modified conditioning distributions on the performance of generated molecules is detailed in Tab.6.

## A.3 FLOW OF ATOM TYPE ON SIMPLEX

Based on Eq.1, $s_i$ undergoes a `softmax` operation, resulting in a normalized probability distribution. The $i$-th dimension of this distribution approaches 1, while the remaining dimensions converge to 0, indicating that the probability mass is correctly concentrated on the atom type $a_i$. Specifically, `softmax`($s_i$) represents a point on the D-category probability simplex $\Delta^{D-1}$, where $\Delta^{D-1} := \left\{ \mathbf{p} \in \mathbb{R}^D : 0 \le \mathbf{p}[i] \le 1, \sum_{i=1}^D \mathbf{p}[i] = 1 \right\}$ is a smooth manifold in $\mathbb{R}^D$ (Wang & Carreira-Perpinán, 2013; Richemond et al., 2022; Floto et al., 2023). Each point on $\Delta^{D-1}$ represents a categorical distribution over $D$ classes. Instead of directly constructing flows on $\Delta^{D-1}$, as detailed in Li et al. (2024), we define a conditional flow in the logit space $\mathbb{R}^D$ with a Gaussian prior, as shown in Eq.2.

During inference, Euler steps are performed to solve the probability flow in the logit space, and atom types can then be sampled from the corresponding probability vector on the simplex:

$$S'_{t+\Delta t} = S_t + v_\theta^{(S)}(t, \mathbf{M}, \mathcal{P})\Delta t, \tag{24}$$

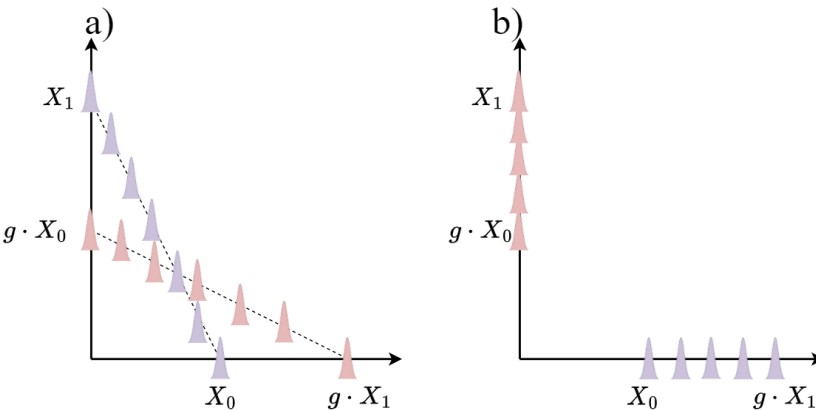

Figure 2: Different sampling methods lead to distinct basic conditional probability paths during training: a) the original path and b) the modified path.

$$A_{t+\Delta t} \sim \texttt{softmax}\left(S'_{t+\Delta t}, dim = 1\right), \tag{25}$$

where $A_{t+\Delta t} \in \mathbb{R}^{N_M \times D}$ denotes atom type sampled according to Eq.25.

To improve consistency between the logit space and the simplex space during generation, the predicted atom types are mapped back to the logit space throughout the iterative process:

$$S_{t+\Delta t} = \texttt{logit}(A_{t+\Delta t}). \tag{26}$$

Additionally, according to Appendix A.1 of Li et al. (2024), a rigorous and correct approach involves reducing one degree of freedom in the logit space to ensure a one-to-one mapping between the simplex and the logit space. However, this setting is not adopted in the implementation of Li et al. (2024), and it does not show a significant impact in PFM.

### A.4 ANALYSIS OF GUIDANCE

For a flow trained with a Gaussian path, if the velocity field can be written in the following form, then the two guidance schemes described in Sec.3.6 can be applied verbatim (Lipman et al., 2024; Zheng et al., 2023):

$$u_t^{\circ}(X) = a_t X + b_t \nabla \log p_t(X), \tag{27}$$

where $a_t$ and $b_t$ are time-dependent coefficients. However, analogous to the discussion in Appendix A.1, it is difficult to express the PFM case in the above form.

From the guidance perspective in Sec.3.6, for classifier guidance one may simply regard $\nabla \log p_{Y|t}^{\phi}(y|\mathbf{M}, \mathcal{P})$ as an additional velocity that points in the direction of increasing the probability of label $y$, and thus use it for guidance. Similarly, for classifier-free guidance, one can view it as adjusting the weights of the guided and unguided velocities along the generative process to an appropriate level to improve generation quality. Therefore, both guidance schemes are directly applicable in our perturbed setting.

## B TRAINING AND SAMPLING PROCEDURE

The training and sampling procedures are summarized as Algorithm 1 and Algorithm 2.

## C IMPLEMENTATION DETAILS

### C.1 INITIALIZATION OF INPUTS

Following Guan et al. (2023a), protein atoms are represented using a combination of one-hot encoding for element types (H, C, N, O, S, Se) and amino acid types (20 types), while ligand atoms

---

**Algorithm 1** Training Procedure of PFM

---

**Input:** Protein-ligand binding dataset $\{\mathcal{P}, \mathcal{M}\}_{i=1}^N$, learnable Flow Matching model $\psi_\theta$, and pretrained perturbed network $\hat{X}$ and $\hat{S}$.

1: **while** $\psi_\theta$ not converge **do**
2:    $(\mathcal{M}_1 = \{(x_i^{\mathcal{M}}, a_i^{\mathcal{M}})\}_{i=1}^{N_M}, \mathcal{P}_1) \sim \{\mathcal{P}, \mathcal{M}\}_{i=1}^N$
3:    $t \sim \mathcal{U}(0, 1)$
4:    Obtain the molecular representation $\mathbf{M}_1 = [X_1, S_1]$ based on **Problem Definition** in Sec.3.2 and Eq.1
5:    Move the complex to make CoM of protein atoms zero
6:    Obtain $\mathbf{M}_0 = [X_0, S_0]$ based on Eq.5 and Appendix A.2
7:    Calculate the perturbation $\hat{X}(S_1, X_0, \mathcal{P}_1)$ and $\hat{S}(X_1, S_0, \mathcal{P}_1)$
8:    $X_t \sim \mathcal{N}(X|tX_1+(1-t)X_0+\xi t(1-t)\hat{X}, \sigma^2 I), S_t \sim \mathcal{N}(S|tS_1+(1-t)S_0+\xi t(1-t)\hat{S}, \sigma^2 I)$

9:    $[\tilde{X}, \tilde{S}] = \psi_\theta(t, X_t, S_t, \mathcal{P}_1)$
10:   Compute loss $\mathcal{L}_{PFM}$ with $[\tilde{X}, \tilde{S}]$ and $[X_1, S_1]$
11:   Update $\theta$ by minimizing $\mathcal{L}_{PFM}$
12: **end while**

---

**Algorithm 2** Sampling Procedure of PFM

---

**Input:** The protein binding site $\mathcal{P}$, the learned Flow Matching model $\psi_\theta$, and pretrained perturbed network $\hat{X}$ and $\hat{S}$.

**Output:** Generated ligand molecule $\mathcal{M}$ that binds to the protein pocket $\mathcal{P}$

1: Sample the number of atoms $N_M$ of the ligand molecule $\mathcal{M}$ as described in **Problem Definition** in Sec.3.2
2: Move CoM of protein atoms to zero
3: Sample initial ligand atom coordinates $X_0$ and continuous atom types $S_0$, forming $\mathbf{M}_0 = [X_0, S_0]$
4: Obtain step size $\Delta t = 1/N_{step}$
5: **for** $n$ in $0, \ldots, N_{step} - 1$ **do**
6:    $[\tilde{X}, \tilde{S}] = \psi_\theta(n\Delta t, X_{n\Delta t}, S_{n\Delta t}, \mathcal{P})$
7:    Use Eq.22 to compute $u_t^\circ$.
8:    $\mathbf{M}_{n\Delta t+\Delta t} = \mathbf{M}_{n\Delta t} + u_t^\circ \Delta t$
9:    Using Eq.25 and Eq.26 to improve consistency between the logit space and the simplex space
10: **end for**
11: $A_1 \sim \texttt{softmax}(S_1, dim = 1)$
12: Employ the method in Guan et al. (2023a) to recover the ligand molecule $\mathcal{M}$ that binds to the protein pocket $\mathcal{P}$

---

are encoded using one-hot representations of their element types (C, N, O, F, P, S, Cl). Furthermore, an additional dimension is introduced to distinguish between protein and ligand atoms. Two linear layers map atoms from different sources to 128-dimensional latent space, respectively. Edge connections of graphs are encoded using a 4-dimensional one-hot vector indicating the connection type (intra-protein, intra-ligand, protein-ligand, or ligand-protein). Edge distances are embedded via radial basis functions located at 20 centers between 0 Å and 10 Å.

## C.2 ARCHITECTURE

An 8-layer SE(3)-equivariant neural network with a hidden dimensionality of 128 and 16 attention heads is used for dot product attention-based message passing. At each layer, three knn graphs ($\mathcal{G}_\mathcal{P}$, $\mathcal{G}_\mathcal{M}$ and $\mathcal{G}_\mathcal{I}$) are dynamically constructed based on distances, where $k_\mathcal{P} = 24$, $k_\mathcal{M} = 32$ and $k_\mathcal{I} = 32$. Message passing and feature updates are conducted over these graphs according to Sec.3.5.

### C.3 TRAINING DETAILS

The model is trained using the Adam optimizer with learning rate of 5e-4, betas = (0.95, 0.999), batch size of 16, and clipped gradient norm of 8. During the training process for $\hat{X}$ predictor and $\hat{S}$ predictor, Gaussian noise is added to $S_1$ and $X_1$ with standard deviations of 2.5 and 0.5, respectively, to adapt to vector field predictions with noise during inference. To ensure appropriate perturbation tendencies of $\hat{X}$, the operations described in Appendix A.2 are applied to $X_0$ and $X_1$. Throughout training, protein atom coordinates are augmented with Gaussian noise with a standard deviation of 0.1. We trained the parameterized PFM on a single NVIDIA GeForce RTX 4090 GPU, and it could converge within 62.5k steps.

### C.4 HYPERPARAMETER SETTINGS

In the experiments, Eq.6 uses $\sigma = 0.1$ and $\xi = 1.5$, while Eq.10 employs $\gamma = 3.5$ and $\mathcal{S}$ uses $\tilde{\sigma} = 2$. In Eq.11, $\mu = 0.75$ and $\nu = 10$. For ablation studies, $\zeta = 2.53$ in LPFM and RPFM to align with the peak values of perturbation terms in the PFM schedule. For guidance, we set $w = 25$ and $\tilde{w} = 5$.

## D ADDITIONAL RESULTS

### D.1 COMPUTATIONAL COST ANALYSIS

Tab.7 presents the training and inference times of PFM compared with two other diffusion-based models, with all models configured according to the settings specified in the respective paper. Training time refers to the duration required for model convergence, while inference time denotes the time taken to generate 10,000 valid molecules. All tests were conducted on the same NVIDIA GeForce RTX 4090 GPU.

Table 7: Computational cost analysis of PFM and baselines, * indicates that the parameters do not include pretrained models.

| Methods | Training | | | Inference | |
|---|---|---|---|---|---|
| | Time(hrs)($\downarrow$) | Parameters(M)($\downarrow$) | GPU Memory Usage(GB)($\downarrow$) | Time(hrs)($\downarrow$) | GPU Memory Usage(GB)($\downarrow$) |
| TargetDiff | 5.57 | 2.82 | 19.98 | 22.11 | 8.51 |
| IPDiff | 9.53 | 2.86* | 16.93 | 66.68 | 21.29 |
| BFM | 4.31 | 7.30 | 18.28 | 1.17 | 6.17 |
| $\hat{X}$ predictor | 4.22 | 7.23 | 18.18 | - | - |
| $\hat{S}$ predictor | 4.17 | 7.23 | 18.18 | - | - |
| PFM | 6.31 | 7.30* | 21.23 | 3.13 | 9.37 |

### D.2 CHOICE OF THE BASIC PATH AND PERTURBATION COEFFICIENTS

Tab.8 reports the impact of basic path choices for atomic coordinates and types on generative performance. **BFM** denotes the choice adopted in this work; **SB** refers to the Schrödinger Bridge path from Tong et al. (2024); **FM** indicates the path from Lipman et al. (2022). For atom types, we also evaluate two additional discrete paths from Campbell et al. (2024) that differ in their initial noise distributions: **mask** concentrates probability mass on a dedicated mask token, whereas **uniform** initializes with a uniform distribution.

Table 8: On the choice of the basic conditional probability path

| Methods | | Vina Score ($\downarrow$) | | Vina Min ($\downarrow$) | | QED ($\uparrow$) | | SA ($\uparrow$) | |
|---|---|---|---|---|---|---|---|---|---|
| | | Avg. | Med. | Avg. | Med. | Avg. | Med. | Avg. | Med. |
| **Coordinate** | BFM | -5.77 | -6.27 | -6.76 | -6.71 | 0.47 | 0.48 | 0.54 | 0.54 |
| | SB | -5.24 | -5.30 | -6.11 | -5.91 | 0.43 | 0.44 | 0.50 | 0.50 |
| | FM | -5.60 | -5.61 | -6.43 | -6.12 | 0.46 | 0.47 | 0.52 | 0.52 |
| **Type** | DFM(mask) | -3.07 | -3.81 | -4.60 | -4.64 | 0.33 | 0.32 | 0.59 | 0.58 |
| | DFM(uniform) | -5.12 | -5.43 | -5.64 | -5.51 | 0.38 | 0.37 | 0.58 | 0.57 |

Tab.9 presents an ablation study on the perturbation coefficient, showing that $\xi = 1.5$ is a generally better choice; $\xi = 0$ corresponds to BFM, while $\xi = 1.5$ corresponds to PFM.

Table 9: Ablation of perturbation coefficients.

| Methods | Vina Score ($\downarrow$) | | Vina Min ($\downarrow$) | | QED ($\uparrow$) | | SA ($\uparrow$) | |
|---|---|---|---|---|---|---|---|---|
| | Avg. | Med. | Avg. | Med. | Avg. | Med. | Avg. | Med. |
| $\xi = 0$ | -5.77 | -6.27 | -6.76 | -6.71 | 0.47 | 0.48 | 0.54 | 0.54 |
| $\xi = 0.5$ | -6.68 | -7.15 | -7.48 | -6.30 | 0.48 | 0.49 | 0.48 | 0.48 |
| $\xi = 1.0$ | -6.78 | -6.83 | -7.25 | -7.13 | 0.48 | 0.49 | 0.52 | 0.52 |
| $\xi = 1.5$ | -7.12 | -7.18 | -7.58 | -7.32 | 0.49 | 0.50 | 0.51 | 0.51 |

### D.3 BROADER VALIDATION OF THE PROPOSED PATH DESIGN

On the SBDD task, we compare with DiffSBDD (Schneuing et al., 2024) on the BindingMOAD dataset (Hu et al., 2005). DiffSBDD's distinctive feature is its diffusion process with pocket noise injection. Data preprocessing follows DiffSBDD, and the experimental results are shown in Tab.10. From these results, PFM achieves better energy (Vina) than DiffSBDD and delivers substantial quality improvements over BFM. PFM's hyperparameters match those used on the CrossDocked2020 dataset.

Table 10: Performance comparison of PFM with DiffSBDD and BFM on BindingMOAD dataset.

| Methods | Vina(All)($\downarrow$) | Vina(Top-10%)($\downarrow$) | QED($\uparrow$) | SA($\uparrow$) | Lipinski($\uparrow$) | Diversity($\uparrow$) | #Atoms/mol. |
|---|---|---|---|---|---|---|---|
| TestSet | $-8.412 \pm 2.03$ | - | $0.522 \pm 0.17$ | $0.692 \pm 0.12$ | $4.669 \pm 0.49$ | - | 28.0 |
| DiffSBDD-cond | $-7.171 \pm 1.89$ | $-9.184 \pm 2.23$ | $0.436 \pm 0.20$ | $0.568 \pm 0.12$ | $4.542 \pm 0.79$ | $0.714 \pm 0.08$ | 24.4 |
| DiffSBDD-joint | $-7.309 \pm 4.03$ | $-9.840 \pm 2.18$ | $0.542 \pm 0.21$ | $0.615 \pm 0.12$ | $4.777 \pm 0.53$ | $0.739 \pm 0.05$ | 25.2 |
| BFM (20steps) | $-6.893 \pm 2.52$ | $-11.566 \pm 1.12$ | $0.410 \pm 0.16$ | $0.628 \pm 0.11$ | $4.716 \pm 0.51$ | $0.881 \pm 0.08$ | 19.7 |
| PFM (20steps) | $-8.649 \pm 3.59$ | $-15.402 \pm 1.54$ | $0.501 \pm 0.15$ | $0.578 \pm 0.15$ | $4.881 \pm 0.36$ | $0.864 \pm 0.09$ | 21.3 |

For molecular generation, we follow SimplexFlow's (Dunn & Koes, 2024) setup on QM9 dataset (Ruddigkeit et al., 2012). Although SimplexFlow is not a particularly strong model, it clearly highlights the gains from perturbation, as shown in Tab.11. Concretely, we apply our perturbation strategy to the SimplexFlow framework by modifying the probability paths of atom types and coordinates, while keeping the original (unperturbed) paths for bond types and atomic charges. When training the predictors, we only adjust the inputs and the coefficients of the loss function.

Table 11: Performance comparison of PerturbedSimplexFlow with SimplexFlow on QM9 dataset.

| Methods | Atoms Stable(%) ($\uparrow$) | Mols Stable(%) ($\uparrow$) | Mols Valid(%) ($\uparrow$) | JS(E) ($\downarrow$) |
|---|---|---|---|---|
| SimplexFlow | 98.0 | 69.7 | 71.9 | 0.82 |
| PerturbedSimplexFlow | 98.2 | 72.7 | 73.8 | 0.33 |

### D.4 VISUALIZATION

Fig.3 visualizes the influence of perturbation in PFM on molecular generation outcomes and processes. Specifically, **PFM (BFM sampling)** refers to molecule generation using only the vector field predictions from the BFM component, while **PFM (BFM part)** represents the BFM component's contribution when generating molecules with the total PFM vector field predictions. **Molecule-level Comparison** of vector fields illustrate the average atomic contributions. The top row of Fig.3 shows performance improvements due to the perturbed conditional probability path, and the bottom two rows demonstrate how perturbations alter the vector fields. The vector field in the figure is not smooth, as it is produced by the trained network rather than the true marginal distribution. Fig.5 provides more visualizations.

Compared with TargetDiff, DecompDiff, and IPDiff, PFM achieves superior or competitive molecular generation results with fewer sampling steps (100 steps in the primary experiment). Fig.4 shows

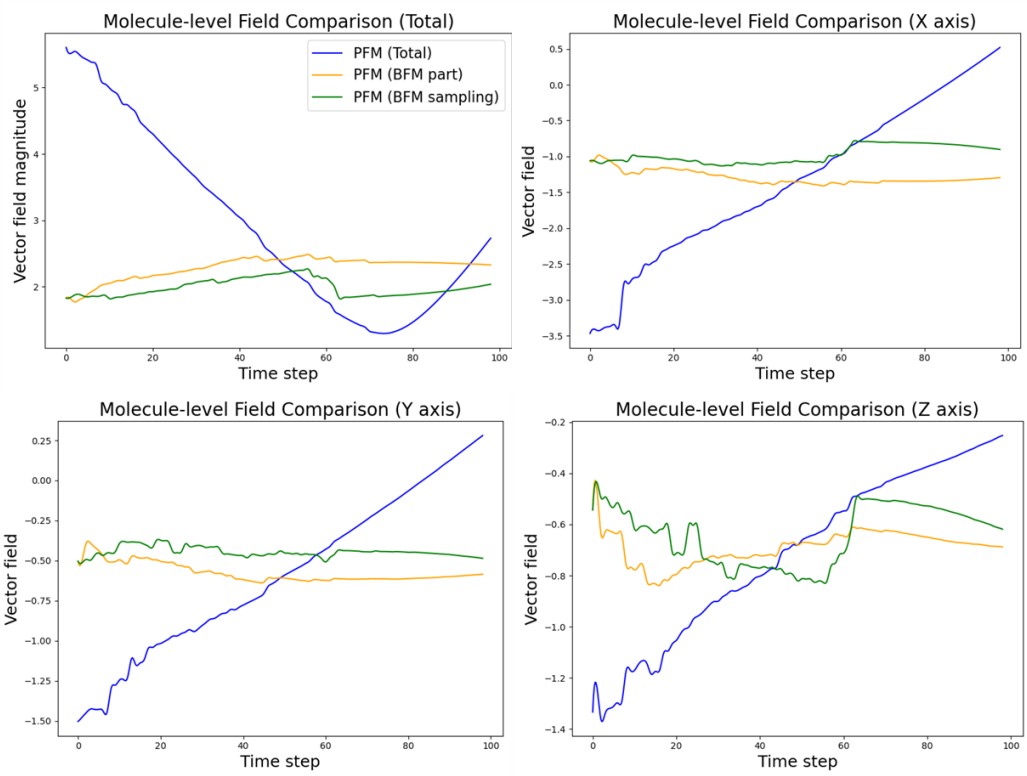

Figure 3: The influence of perturbation in PFM on molecular generation outcomes (top row) and processes (middle and bottom rows) on protein 2v3r.

that further reducing the number of sampling steps does not lead to significant fluctuations in the quality of generated molecules. This indicates the method's potential for faster molecule generation.

In addition, we provide the visualization of more ligand molecules generated by PFM, comparing with both reference and IPDiff (Huang et al., 2024), as shown in Fig.6.

USE OF LLMS

We used LLMs to polish the English writing.

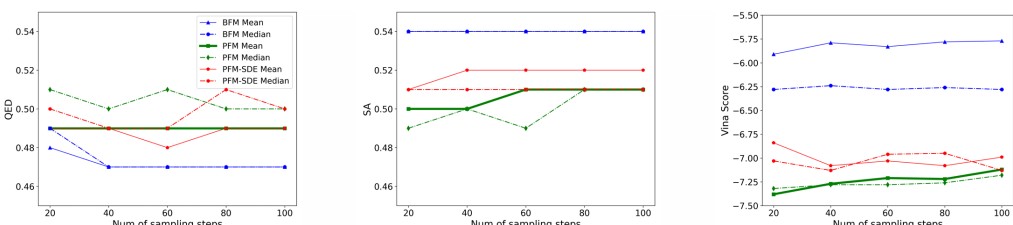

Figure 4: Ablation study on sampling step number. We compare PFM with BFM in terms of QED, SA, Vina Score under different sampling step number settings.

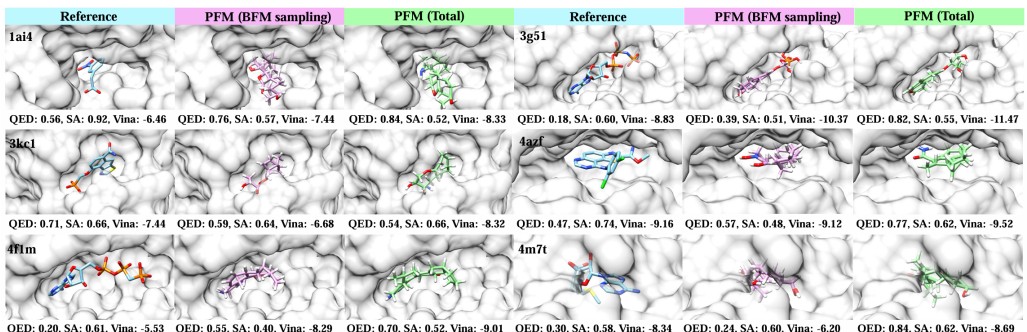

Figure 5: More examples of generated ligands for protein pockets. The generated molecules are produced by the trained PFM model. The results of PFM (BFM sampling) and PFM (Total) share the same noise initialization but differ in the vector field predictions used during the generation process.

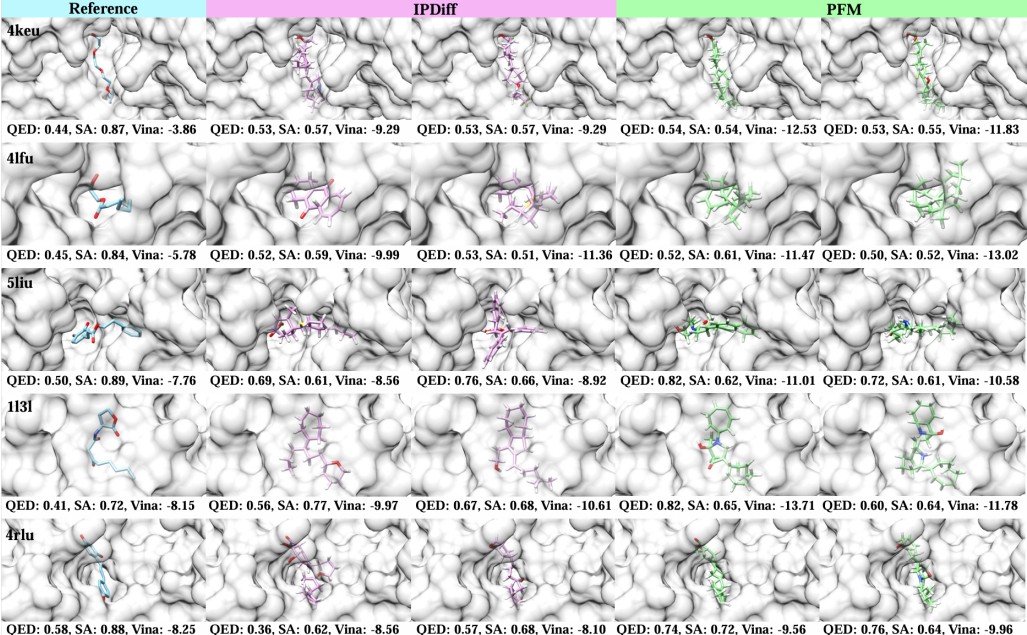

Figure 6: More examples of generated ligands for protein pockets. Carbon atoms in reference ligands, ligands generated by IPDiff (Huang et al., 2024) and PFM are visualized in blue, pink and green, respectively. QED, SA and Vina Score are reported for each molecule.

