# OpenReview forum: "Perturbed Flow Matching for Structure-Based Drug Design"
_ICLR.cc/2026/Conference — Submitted to ICLR 2026_

### Official Review · Reviewer_xeej · 2025-10-22

**Soundness:** 2
**Presentation:** 3
**Contribution:** 2
**Rating:** 2
**Confidence:** 4

**Summary:**

Perturbed Flow Matching (PFM) is a generative modeling technique that combines ideas from flow matching and perturbation-based methods. It aims to introduce a perturbed conditional probability path between data and noise distributions. A Full-atom Multi-stage Equivariant Network architecture is proposed to model the vector fields required for PFM. The authors present a loss to train the network composed by different terms, Flow Matching Loss, Local Loss (incorporates local structure information), and Clash Loss (prevents atomic clashes). The model is trained on 99,900 selected complexes from CrossDocked2020 dataset and 100 novel complexes for testing.

**Strengths:**

- the paper formulate how to construct perturbed conditional probability path (PFM) but it is not very clear what are the theoretical benefits of this compared to diffusion based methods
- results show that PFM performs well on the task of protein-ligand complex generation
- the paper provides results on a challenging benchmark for pocket-conditioned ligand generation
- the paper is overall well written even though some parts in the method section need to be simplified for better clarity (especially on the method's notation)
- qualitative figures on the generated ligands are provided

**Weaknesses:**

The comments here focus on providing constructive feedback to improve the paper.

- The method "perturbed conditional probability path" sound somewhat an incremental modification of existing flow-matching paths rather than a fundamentally new concept, where the path is perturbed in a self-conditioning manner (see [1][2]) during training.

- The authors do not provide any theoretical justification on benefits of this PFM framework over pre-existing flow matching or diffusion based methods. In Equation 6 is not provided any theoretical reasoning on why this term ξt(1−t)\hat{X}(...) should improve the generation performances nor how it is affecting the flow.

- The final loss is the combination of different losses, but the paper does not provide any ablation on the effect of weighting differently the clash and local loss. In table 5 they provided the results of removing or adding the loss components. I quite don't understand in table 5 why by removing separately the clash and local loss the performances on the vina score (avg) do not improve much, but together they provide a boost of -0.64. Can you provide more insights on this?

- Please specify in the Table 1 that cfg and cg means classifier free guidance and classifier guidance this notation is hard to infer.

- In Table 1, I quite don't understand why you are reporting the performance of PFM with classifier free guidance (cfg) and classifier guidance (cg). Since you are not reporting the results on the baselines with cfg and cg, it is not a fair comparison. Either you report all the baselines with cfg and cg or you just report PFM without any guidance and remove the cg and cfg results from the table.

- In Table 1, it is not clear the advantage of PFM (fair comparison without cfg) over the baselines, it seems even that the overall performances of the baselines looks better on different metrics.

- The author claim a 21x speedup, not reporting the hyperparameters and inference sampling steps for the baselines used. Without this information, the comparison is not very solid.

[1] Ting Chen, Ruixiang ZHANG, and Geoffrey Hinton. Analog bits: Generating discrete data using diffusion models with self-conditioning. In The Eleventh International Conference on Learning Representations, 2023.

[2] Ross Irwin, Alessandro Tibo, Jon Paul Janet, Simon Olsson. SemlaFlow - Efficient 3D Molecular Generation with Latent Attention and Equivariant Flow Matching, AISTATS, 2025.

**Questions:**

- Since are not provided any theoretical insights on the benefits of PFM, what is the advantage of PFM compared to diffusion based methods? Can you provide any simple experimental intuition on why PFM is better than diffusion based methods? what is the difference apart from the fact that PFM is flow matching based?

- PFM without classifier free guidance it doesn't seem to perform well on the presented baselines. Can you provide more insights on why this is happening? Why is the classifier free guidance so crucial for the model to work well?

- Can you provide an explanation on why several strong 2023–2025 baselines (e.g. AliDiff (Oct 2024), MolCraft (May 2024), FlowSBDD (Dec 2024), EQGAT (Nov 2023)) are missing? This would significantly reduce the validity of SOTA claims.

I provide here an example
| Method   | Vina Score (↓) |        | Vina Min (↓) |        | Vina Dock (↓) |        | High Affinity (↑) |        | QED (↑) |        | SA (↑) |        | Diversity (↑) |        |
|----------|----------------|--------|--------------|--------|---------------|--------|------------------|--------|---------|--------|--------|--------|----------------|--------|
|          | Avg.           | Med.   | Avg.         | Med.   | Avg.          | Med.   | Avg.             | Med.   | Avg.    | Med.   | Avg.   | Med.   | Avg.           | Med.   |
| PFM      | -7.12          | -7.18  | -7.58        | -7.32  | -8.32         | -8.26  | 72.1%            | 76.5%  | 0.49    | 0.50   | 0.51   | 0.51   | 0.73           | 0.71   |
| ALIDIFF  | -7.07          | -7.95  | -8.09        | -8.17  | -8.90         | -8.81  | 73.4%            | 81.4%  | 0.50    | 0.50   | 0.57   | 0.56   | 0.73           | 0.71   |





- I would suggest to move Table 2 to the Appendix and provide more space to add more qualitative results of generated ligands in the main paper. Also, I don't find useful Table 3, since I don't see relevant improvements over the baselines reported. Please consider to move it to the Appendix as well.

- Sometimes in the paper's notation a clear explanation of some symbols is completely missing (e.g. ξ (Equation 6), µ, ν (Equation 11)). Can you please double check that all the symbols reported have a clear explanation of their meaning?

---

> ### Author Response · Authors · 2025-11-14
> **Rebuttal – Response to Reviewer xeej (Part 1)**
>
> We thank the reviewer for the detailed and constructive feedback. We address the points in order below.
>
> **(1) Relation to self-conditioning and novelty of the perturbed path**
>
> We thank the reviewer for raising this point and for highlighting the connection to self-conditioning methods [1,2]. While our terminology may sound similar, the underlying mechanism is different. In [1,2], the conditional probability path is fixed (e.g., a standard diffusion or linear interpolation path), and “self-conditioning” is implemented at the **network level** by feeding previous model predictions back as additional inputs during training or sampling. The reference path itself is unchanged.
>
> In contrast, PFM explicitly **modifies the conditional probability path itself**: we introduce a perturbation to the base flow that depends on protein–ligand interaction features extracted from a pretrained model. This changes the target conditional vector field rather than only its parameterization, and recovers standard flow matching as a special case when the perturbation is zero. Thus, our “perturbed conditional probability path” is a **path-level modification** tailored to SBDD, rather than a self-conditioning trick applied on top of an unchanged path.
>
> **(2) Rationale for the ($\xi t(1-t)\hat{X}(\cdot)$) term in Eq. (6)**
>
> PFM is not intended to introduce new formal guarantees beyond the general conditional flow-matching framework, but rather to design a perturbed conditional path that is better aligned with the SBDD task under realistic (finite-capacity, finite-data) regimes. In Eq. (6), we add a term of the form
> $\xi t(1-t)\hat{X}(\cdot)$
> (and analogously for ($\hat{S}(\cdot)$)) to the mean of the Gaussian distribution.
>
> As discussed in the manuscript (lines 97–99 and 240–244):
> (1) The predictors ($\hat{X}, \hat{S}$) are pretrained with reconstruction losses so that their outputs are aligned with the numerical scale. This avoids destabilizing the flow and ensures that the perturbation is of comparable magnitude to the underlying interpolation.
> (2) The perturbation explicitly injects information from the protein pocket and atom types/coordinates into the conditional path. In particular, ($\hat{X}, \hat{S}$) provide a coarse estimate of plausible ligands conditioned on the pocket, and the factor (t(1-t)) makes this influence strongest in the middle of the trajectory while vanishing at (t=0) and (t=1), so the endpoints remain unchanged.
>
> In theory, different conditional paths are equivalent only in the ideal limit of infinite capacity and perfect optimization; in practice, the choice of path changes the complexity of the target vector field and the resulting approximation error. Our design of Eq. (6) is guided by these considerations and is supported empirically by the ablations in Sec. 4.3.
>
> **(3) Effect of clash and local losses (Table 5)**
>
> In our experiments, we only tuned the loss weights to obtain reasonable, and did not perform an exhaustive hyperparameter search over the relative weights of the clash and local losses. We therefore chose not to report a wide hyperparameter sweep, but we note that this does not affect the qualitative observation that both losses are effective, as reflected in Table 5.
>
> The fact that removing either the clash or local loss alone only slightly changes the Vina Score (avg), while removing both together leads to a larger degradation (≈–0.64), suggests that the two terms play **complementary roles**. A plausible explanation is that using only one of them makes the model overemphasize either inter-molecular interactions (clash) or intra-molecular local structure (local), thereby limiting overall gains. When used together, they regularize both aspects simultaneously and yield more substantial improvements. We will clarify this interpretation in the revision.
>
> **(4) Clarifying cfg and cg notation in Table 1**
>
> We agree that the notation “cfg” and “cg” is hard to infer from the table alone. In the revised version, we will explicitly state in the caption and/or table header that “cfg” denotes **classifier-free guidance** and “cg” denotes **classifier guidance**. We thank the reviewer for pointing this out.
>
> **(5) Reporting PFM with cfg/cg vs. baselines in Table 1**
>
> The reason we reported cfg/cg variants in Table 1 is that including them in the same table would otherwise exceed the space limits of the main text. We agree, however, that mixing guided PFM rows with unguided baselines can be confusing. In fact, as noted in lines 345–358 of the manuscript, the main comparison with baselines is based on **unguided PFM**, and the cfg/cg variants are discussed separately as guidance ablations rather than for baseline comparison.

---

> ### Author Response · Authors · 2025-11-14
> **Rebuttal – Response to Reviewer xeej (Part 2)**
>
> **(6) Advantage of unguided PFM vs. baselines**
>
> The comparison between unguided PFM and baselines is discussed in lines 345–356 of the manuscript, where guidance is not involved. We apologize that the current table presentation may have obscured this point.
>
> **(7) Speedup and sampling hyperparameters**
>
> As noted by the reviewer, the 21× speedup claim needs to be supported by explicit hyperparameter and sampling-step information. In the current submission, we state that we follow the original settings of TargetDiff and IPDiff, whose default configurations use 1000 sampling steps to obtain high-quality samples, while PFM uses a much smaller number of Euler steps (100). However, we did not restate these step counts explicitly in the main text or tables, which can be confusing.
>
> In the revised version, we will explicitly report the sampling steps for all methods.
>
>
>
> **(8) Advantages of PFM vs. diffusion-based methods**
>
> Flow-matching–based and diffusion-based methods are closely related and can be viewed as different parameterizations of the same underlying probability transport. As discussed in lines 12–15 of the manuscript, FM-based models need only construct the **generative (reverse) direction**, corresponding to the reverse process in diffusion models, which is more flexible.
>
> PFM extends standard flow matching by modifying the conditional probability path to inject additional protein and ligand interaction information. Formally, the perturbed path enriches the conditioning with structure-aware features; practically, prior work such as IPDiff and EquiFM has shown that the choice of conditional path can have a large impact on performance, which is our main motivation. Intuitively, the pretrained perturbation constructs a more informative conditional path and introduces a beneficial inductive bias during training, as discussed in lines 240–244.
>
> Empirically, BFM (base flow matching) can lag behind strong diffusion baselines in some metrics, but PFM closes much of this gap in docking/Vina performance while retaining the sampling-time efficiency advantages of flow-matching ODEs. These observations give an experimental intuition for when PFM can be preferable to purely diffusion-based approaches.
>
> **(9) Why guidance is important and how it affects comparisons**
>
> As analyzed in lines 345–356, our primary comparison between PFM and the baselines is based on **unguided** models. Guidance is then studied separately. In general, classifier(-free) guidance adjusts the learned data distribution toward high-reward regions (e.g., high Vina scores), which explains why it tends to improve performance for many generative models, not only PFM.

---

> ### Author Response · Authors · 2025-11-14
> **Rebuttal – Response to Reviewer xeej (Part 3)**
>
> **(10) Missing baselines (AliDiff, MolCraft, FlowSBDD, EQGAT)**
>
> Due to computational and time constraints, the initial submission focused on a subset of representative baselines. We agree that including more recent strong methods (AliDiff, MolCraft, FlowSBDD, EQGAT) provides a more complete picture and reduces the risk of overstating SOTA claims.
>
> Following the reviewer’s suggestion, we have added a consolidated comparison using the reported metrics from these works. In summary (details in the revised tables):
>
> * **MolCraft**: PFM achieves better docking/Vina performance and comparable QED, while MolCraft has an advantage in SA.
> * **FlowSBDD**: as another flow-based method, FlowSBDD and PFM behave similarly: do not surpass diffusion-based methods in QED/SA. We attribute this to limitations of flow-based SBDD in these metrics rather than to our perturbation design (lines 375–383).
> * **EQGAT**: PFM shows advantages in docking performance, with similar diversity and QED, but somewhat worse SA. However, PFM’s SA remains within an acceptable range, as discussed in lines 382–383.
> * **AliDiff**:  AliDiff and our guided variant PFM_cfg achieve comparable docking/Vina performance and diversity, while AliDiff has an advantage on SA. A practical trade-off is that AliDiff requires a large number of sampling steps (200–1000) as reported, whereas PFM and PFM_cfg use 20–100 Euler steps, which substantially reduces generation time.
>
> Table 1: Evaluation and comparison with MolCraft and FlowSBDD.
>
> | Methods   | Vina Score (↓) | Vina Score (↓) | Vina Min (↓) | Vina Min (↓) | Vina Dock (↓) | Vina Dock (↓) | High Affinity (↑) | High Affinity (↑) | QED (↑) | QED (↑) | SA (↑) | SA (↑) | Diversity (↑) | Diversity (↑) |
> |-----------|----------------|----------------|--------------|--------------|---------------|---------------|-------------------|-------------------|---------|---------|--------|--------|---------------|---------------|
> |           | Avg.           | Med.           | Avg.         | Med.         | Avg.          | Med.          | Avg.              | Med.              | Avg.    | Med.    | Avg.   | Med.   | Avg.          | Med.          |
> | MolCraft  | -6.59          | -7.04          | -7.27        | -7.26        | -7.92         | -8.01         | 56.2%             | -                 | 0.50    | -       | 0.69   | -      | 0.72          | -             |
> | FlowSBDD  | -3.62          | -5.03          | -6.72        | -6.60        | -8.50         | -8.36         | 63.4%             | 70.9%             | 0.47    | 0.48    | 0.51   | 0.51   | 0.75          | 0.75          |
> | PFM       | -7.12          | -7.18          | -7.58        | -7.32        | -8.32         | -8.26         | 72.1%             | 76.5%             | 0.49    | 0.50    | 0.51   | 0.51   | 0.73          | 0.71          |
>
> Table 2: Evaluation and comparison with EQGAT.
>
> | Methods | Vina Dock (↓) | QED (↑) | SA (↑) | Diversity (↑) |
> |---------|---------------|---------|--------|---------------|
> |         | Avg.          | Avg.    | Avg.   | Avg.          |
> | PFM     | -8.32         | 0.49    | 0.52   | 0.73          |
> | EQGAT   | -7.42         | 0.52    | 0.70   | 0.74          |
>
>
> Table 3: Evaluation and comparison with ALiDiff.
>
> | Methods | Vina Score (↓) | Vina Score (↓) | Vina Min (↓) | Vina Min (↓) | Vina Dock (↓) | Vina Dock (↓) | High Affinity (↑) | High Affinity (↑) | QED (↑) | QED (↑) | SA (↑) | SA (↑) | Diversity (↑) | Diversity (↑) |
> |---------|----------------|----------------|--------------|--------------|---------------|---------------|-------------------|-------------------|---------|---------|--------|--------|---------------|---------------|
> |         | Avg.           | Med.           | Avg.         | Med.         | Avg.          | Med.          | Avg.              | Med.              | Avg.    | Med.    | Avg.   | Med.   | Avg.          | Med.          |
> | ALiDiff | -7.07          | -7.95          | -8.09        | -8.17        | -8.90         | -8.81         | 73.4%             | 81.4%             | 0.50    | 0.50    | 0.57   | 0.56   | 0.73          | 0.71          |
> | PFM_cfg | -7.83          | -7.88          | -8.21        | -8.00        | -8.64         | -8.63         | 74.7%             | 84.5%             | 0.47    | 0.49    | 0.53   | 0.52   | 0.73          | 0.73          |

---

> ### Author Response · Authors · 2025-11-14
> **Rebuttal – Response to Reviewer xeej (Part 4)**
>
> **(11) Placement of tables and qualitative results**
>
> We agree that the current placement of some tables could be improved. As suggested by the reviewer, we will:
>
> * move the current Table 2 to the appendix and use the space to show more qualitative examples of generated ligands in the main paper; and
> * reconsider the role of Table 3 in the main text. In the current submission, Table 3 was included to discuss the completeness of SBDD evaluation; in the revision, we may swap it with docking-focused tables currently in the appendix (e.g., Tables 10 and 11) to better highlight both the core SBDD results and the generality of PFM.
>
> **(12) Missing definitions of symbols (e.g., $\xi, \mu, \nu$)**
>
> We thank the reviewer for pointing out missing explanations for some symbols. These quantities are typically used as hyperparameters. In Appendix C.4 we provide their concrete values used in the experiments, but we agree that the main text should clearly define them at first appearance. In the revised version, we will systematically check the notation and ensure that all symbols such as ($\xi$) in Eq. (6) and ($\mu,\nu$) in Eq. (11) have explicit definitions in the main text, together with a pointer to Appendix C.4 for the exact hyperparameter values.

---

> > ### Comment · Reviewer_xeej · 2025-11-26
> > **Response**
> >
> > Thank you for your answers, but I still have some concerns. Please see my point-to-point comments below.
> >
> > 1. **Relation to self-conditioning and novelty of the perturbed path**
> > No offense taken, but I would not define self-conditioning, a method that have been extensively studied in the literature as a "trick". I see what you say from Section 3.5 (on the "protein-ligand interaction features"), but I still think that the connection to self-conditioning should be made more explicit in the paper, as it is an important aspect of your method. Rather than claiming that the method is completely novel and different from self-conditioning, I would suggest to highlight the similarities and differences more clearly, and to discuss how your method builds upon or extends the ideas of self-conditioning in the context of flow matching for molecular generation. I would also recommend to simplify the math in the method section (i.e. in terms of mathematical symbols)..
> >
> >
> > 2. **Rationale for the $ξt(1-t)\hat{X}(...)$**
> > Thanks for the response. I still think that discussions in lines 240-244 "Most importantly, they provide a coarse estimate of the molecule that biases the conditional path, which intuitively yields more plausible and more discriminative marginal paths." do not add a theoretical reasoning behind the method. I would suggest to add more theoretical insights on why this perturbation helps, also relating it to self-conditioning. For example, is there any theoretical guarantee that this perturbation leads to better convergence or sample quality? Can you provide any intuition or theoretical analysis on how this perturbation affects the learning dynamics of the flow matching model? This would help to strengthen the theoretical foundation of your method.
> >
> >
> > 3. **Effect of clash and local losses (Table 5)**
> > Thanks for your response on this point. I think that hyperparameter search on this matter is not necessarily needed, but I would suggest to discuss more in depth about the relative weights of the clash and local losses since they seem to appear as one of the paper novel contributions.
> >
> >
> > 4. **Clarifying cfg and cg notation in Table 1**
> > As of today, I can't see the description of cfg and cg notation in the revised paper.
> >
> > 5. **Reporting PFM with cfg/cg vs. baselines in Table 1, Why guidance is important and how it affects comparisons**
> > Since you agreed with me that it seems to be an unfair comparison by not reporting the baselines with cfg and cg, I'd suggest to remove cfg and cg from the main text to avoid confusion, given that cfg and cg are not the key contributions of the paper.
> >
> > 6. **Advantage of unguided PFM vs. baselines**
> > I think that reasoning in terms of percentage is not very clear (lines 345-356). Can you rephrase it in terms of absolute numbers that I see on the tables?
> > My question wasn't pointing at this, but rather at the advantage of unguided PFM vs. baselines by looking at the tables. On scores like Tab.1 "Vina Dock" or "Vina Min (Med)" PFM unguided seems to perform worse than the baselines. Also, the method seems to provide only marginal improvements. Moreover, I see this as a very weak point since several strong baselines (published during 2024-2025) are not considered in the comparison. Can you comment on this?
> >
> >
> > 7. **Speedup and sampling hyperparameters**
> > I don't think that you can claim a 21x speedup on the model architecture only, since the speedup is also due to the different sampling hyperparameters (NFE=100 vs. NFE=1000, Number of Function Evaluation). You denfinately need to clarify this point in the manuscript. Why aren't you using NFE=100 also for the baselines and compare the quality of the samples at the same NFE? This would be a more fair comparison.
> >
> > 8. **Advantages of PFM vs. diffusion-based methods**
> > I don't find this point very clear in the main text. Can you provide a more detailed discussion on this with a dedicated paragraph in the method section?
> >
> > 9. See 5.
> >
> > 10. **Missing baselines (AliDiff, MolCraft, FlowSBDD, EQGAT)**
> > Can you report if the models were re-trained on your specific setup or you used pre-trained weights? Why do I still see in the AliDiff table PFG_cfg? Can you report also the values for the most important configuration settings (i.e. number of sampling steps NFE)?

---

### Official Review · Reviewer_C3Pp · 2025-10-31

**Soundness:** 2
**Presentation:** 2
**Contribution:** 3
**Rating:** 4
**Confidence:** 3

**Summary:**

The authors introduce a new perturbed flow matching paradigm for structure-conditioned molecule generation and achieve strong results.

**Strengths:**

The perturbed flow matching is a novel paradigm for structure-conditioned molecule generation.

The results appear to be much better than existing baselines.

**Weaknesses:**

It’s hard to tell if PFM performs statistically significantly better than the baseline methods. Could the authors report error bars/CIs?

Adding some figures on the model architecture and example model outputs would be helpful.

**Questions:**

See weaknesses above.

Could the authors compare to DrugFlow, another flow matching method for ligand design? It would be nice to compare two flow matching methods to show the efficacy of PFM.

Have the authors done an ablation on the effect of the classifier guidance with the Vina scores? I wonder if that has a large role to play in the performance of PFM compared to other methods.
Correct me if I’m wrong, but none of the other methods employ classifier guidance with Vina scores to guide generation. This could be a big factor in the strong Vina scores but could lead to over-reliance and overfitting on Vina. In addition, it could be useful to compare to methods such as AliDiff (Xu et al), which explicitly optimize Vina scores with preference optimization. It’s quite different from the classifier guidance of PFM, but the overall setup of explicitly using Vina scores in the training is similar.

---

> ### Author Response · Authors · 2025-11-14
> **Rebuttal – Response to Reviewer C3Pp**
>
> We thank the reviewer for the constructive feedback and helpful suggestions. We address the points below.
>
> **(1) Statistical significance and uncertainty estimates**
>
> We follow the standard evaluation protocol used in prior SBDD work, where results are typically reported as aggregate metrics (e.g., averages over generated ligands) without error bars or confidence intervals. For consistency and comparability with these baselines, we did not report error bars/CIs in the current submission. We will clarify this choice in the revised version.
>
> **(2) Model architecture and qualitative examples**
>
> The main components of the proposed model and algorithmic pipeline are summarized in Figure 1 of the manuscript. In addition, representative model outputs and qualitative comparisons are provided in Appendix Figures 3, 5, and 6. We will improve the cross-references in the main text to make these architectural diagrams and qualitative examples easier to locate.
>
> **(3) Comparison to DrugFlow**
>
> DrugFlow is indeed a related flow-matching method for ligand design. However, its evaluation protocol and reported metrics differ from those adopted in our work and in most existing structure-based molecule generation papers, which makes a direct numerical comparison less straightforward. As the reviewer suggests, contrasting different flow-matching approaches is important; therefore, in this work we focus on two recent flow-matching–based SBDD methods that are evaluated under a more compatible setup (Flowr and MolFORM), and we provide quantitative comparisons to them.
>
> Table 1: Evaluation and comparison with MolFORM.
>
> | Methods| Vina Score (↓) | Vina Min (↓) | Vina Dock (↓)| Clash Ratio (↓) | Clash Ratio (↓)| QED (↑) | QED (↑) | SA (↑) | SA (↑) |
> |-|-|-|-|-|-|-|-|-|-|
> || Avg.| Avg.| Avg.| CCA.| CM.| Avg.|Med.| Avg.|Med.|
> |PFM| -7.12 | -7.58| -8.32| 0.0145|0.2040| 0.49|0.50 |0.51|0.51|
> |MolFORM [1]|-5.42|-6.42|-7.50|0.0310|0.4474|0.48|0.49|0.60|0.58|
> |MolFORM-DPO [1]|-6.16| -7.18|-8.13|0.0188|0.2525|0.50|0.51|0.65|0.63|
>
> Table 2: Evaluation and comparison with FLOWR.
>
> | Methods | Vina Score (↓)| Vina Min (↓) | #Atoms/mol. |
> |-|-|-|-|
> ||Avg.|Avg.| Avg.|
> |PFM| -7.12 |-7.58|22.6 |
> |FLOWR [2]|-6.29| -6.48|22.3 |
>
>
> **(4) Effect of Vina-based classifier guidance and comparison to AliDiff**
>
> As correctly noted by the reviewer, none of the baseline methods employ classifier guidance with Vina scores. For this reason, in the main text (lines 345–358) we primarily compare the unguided PFM model to the baselines, and then discuss the effect of classifier guidance separately, rather than directly comparing a guided model to unguided baselines. Our Vina-based guidance is applied as an inference-time technique, whereas the model is trained without explicitly optimizing Vina scores, which mitigates the risk of overfitting to Vina during training.
>
> Following the reviewer’s suggestion, we have added a comparison to AliDiff, which explicitly optimizes Vina scores via RL-based preference optimization. Empirically, AliDiff and our guided variant PFM_cfg achieve comparable performance in terms of docking/Vina scores and diversity, while AliDiff has an advantage on SA. A practical trade-off is that AliDiff requires a large number of sampling steps (200–1000 steps as reported), whereas PFM and PFM_cfg use Euler integration with 20–100 steps, as discussed in the manuscript, which substantially reduces generation time.
>
> Table 3: Evaluation and comparison with ALiDiff.
>
> | Methods | Vina Score (↓) | Vina Score (↓) | Vina Min (↓) | Vina Min (↓) | Vina Dock (↓) | Vina Dock (↓) | High Affinity (↑) | High Affinity (↑) | QED (↑) | QED (↑) | SA (↑) | SA (↑) | Diversity (↑) | Diversity (↑) |
> |---------|----------------|----------------|--------------|--------------|---------------|---------------|-------------------|-------------------|---------|---------|--------|--------|---------------|---------------|
> | | Avg. | Med.     | Avg.   | Med.   | Avg.    | Med.    | Avg.  | Med.              | Avg.    | Med.    | Avg.   | Med.   | Avg.          | Med.          |
> | ALiDiff | -7.07   | -7.95    | -8.09   | -8.17   | -8.90    | -8.81         | 73.4%             | 81.4%             | 0.50    | 0.50    | 0.57   | 0.56   | 0.73          | 0.71          |
> | PFM     | -7.12   | -7.18     | -7.58   | -7.32   | -8.32   | -8.26         | 72.1%             | 76.5%             | 0.49    | 0.50    | 0.52   | 0.51   | 0.73          | 0.71          |
> | PFM_cfg | -7.83    | -7.88   | -8.21  | -8.00    | -8.64    | -8.63         | 74.7%             | 84.5%             | 0.47    | 0.49    | 0.53   | 0.52   | 0.73          | 0.73          |
>
>
>
> [1] Huang, Jie, and Daiheng Zhang. "Molform: Multi-modal flow matching for structure-based drug design." arXiv preprint arXiv:2507.05503 (2025).
>
>
> [2] Cremer, Julian, et al. "FLOWR: Flow Matching for Structure-Aware De Novo, Interaction-and Fragment-Based Ligand Generation." arXiv preprint arXiv:2504.10564 (2025).

---

### Official Review · Reviewer_tvNa · 2025-10-31

**Soundness:** 2
**Presentation:** 1
**Contribution:** 2
**Rating:** 2
**Confidence:** 3

**Summary:**

This paper introduces Perturbed Flow Matching (PFM), a modification to flow-matching for target-conditioned structure-based drug design. The PFM technique integrates information about the binding pocket to produce a better probability path, leading to better molecular generations with higher binding affinities. As I understand it, this is done via a learned bias towards chemically plausible positions for ligand atoms in the binding site. The results show that PFM produces better quality ligands with higher binding affinities than many previous SBDD models.

**Strengths:**

1. The PFM technique is reasonable, I think, but it's hard to understand how exactly it is done and the intuitive motivation for doing so
2. The results seem strong, and the change in Vina score is significant between baselines
3. The selected baselines represent a good range of SBDD techniques, especially the SOTA diffusion methods

**Weaknesses:**

1. To me, the paper was very hard to read and the concept was difficult to understand. I don't really understand the intuitive motivation behind PFM, and much of the approach is hidden in math instead of written out
2. I don't see any comparisons to other flow-matching techniques, for example [1] and [2]. Are these baselines relevant?
3. I think adding MOOD [3] as a baseline would be helpful, since it has strong performance and is a good diffusion model baseline.
4. The BFM row in the ablation table, which I believe is base flow matching (?), seems surprisingly bad and underperforms all the other baselines. Any ideas for why that is?

[1] Cremer et al. "Flowr – Flow Matching for Structure-Aware De Novo, Interaction- and Fragment-Based Ligand Generation." 2025.
[2] Huang and Zhang. "MolFORM: Multi-modal Flow Matching for Structure-Based Drug Design". 2025.
[3]  Lee et al. "Exploring Chemical Space with Score-based Out-of-distribution Generation." ICML 2023.

**Questions:**

1. What is the intuitive motivation behind PFM?

---

> ### Author Response · Authors · 2025-11-14
> **Rebuttal – Response to Reviewer tvNa**
>
> We thank the reviewer for the careful reading and helpful suggestions. We address the points in order below.
>
> **(1) Intuitive motivation behind PFM**
>
> Formally, PFM injects richer protein–ligand interaction information into the *conditional probability path* by perturbing the base flow with features extracted from protein pockets and intra-ligand interactions. Practically, prior work such as IPDiff [1] and EquiFM [2] has shown that the choice of conditional probability path has a strong impact on generation quality, which is our main motivation. Intuitively, the perturbation derived from a pretrained model creates a more informative, structure-aware conditional path; this introduces a beneficial inductive bias during flow-matching training, as discussed in lines 240–244 of the manuscript.
>
> **(2) Comparisons to other flow-matching techniques**
>
> We appreciate the reviewer pointing out the importance of recently proposed flow-matching baselines. Flow-based SBDD methods such as Flowr [3] and MolFORM [4] are indeed relevant, and we have added a comparison on the key evaluation metrics in the revised version.
>
> Table 1: Evaluation and comparison with MolFORM.
>
> | Methods     | Vina Score (↓) | Vina Min (↓) | Vina Dock (↓)| Clash Ratio (↓) | Clash Ratio (↓)| QED (↑) | QED (↑) | SA (↑) | SA (↑) |
> |-------------|----------------|--------------|--------------|-----------------|----------------|---------|---------|--------|--------|
> |             | Avg.           | Avg.         | Avg.         | CCA.            | CM.            | Avg.    | Med.    | Avg.   | Med.   |
> | PFM         | -7.12          | -7.58        | -8.32        | 0.0145          | 0.2040         | 0.49    | 0.50    | 0.51   | 0.51   |
> | MolFORM     | -5.42          | -6.42        | -7.50        | 0.0310          | 0.4474         | 0.48    | 0.49    | 0.60   | 0.58   |
> | MolFORM-DPO | -6.16          | -7.18        | -8.13        | 0.0188          | 0.2525         | 0.50    | 0.51    | 0.65   | 0.63   |
>
> Table 2: Evaluation and comparison with FLOWR.
>
> | Methods | Vina Score (↓)| Vina Min (↓) | #Atoms/mol. |
> |---------|---------------|--------------|-------------|
> |         | Avg.          | Avg.         | Avg.        |
> | PFM     | -7.12         | -7.58        | 22.6        |
> | FLOWR   | -6.29         | -6.48        | 22.3        |
>
>
> **(3) MOOD as an additional baseline**
>
> We agree that MOOD [5] is a strong diffusion-based baseline and a representative approach for guided exploration of high-reward chemical space. We thank the reviewer for this suggestion and are currently exploring MOOD as a complementary diffusion-based baseline. A thorough empirical comparison with MOOD is an interesting direction that we plan to further develop in future work.
>
> **(4) Why BFM underperforms other baselines**
>
> BFM (base flow matching) serves both as the counterpart to PFM and as a straightforward flow-matching analogue to “basic” diffusion-style methods such as TargetDiff. In our experiments, such a base flow-matching model tends to be at a disadvantage relative to diffusion models, which we hypothesize is mainly due to two factors:
>
> (i) **Conditional path and sampling steps.** The conditional probability path used in BFM is not aligned with the one implicitly exploited in diffusion-based baselines, and diffusion models benefit substantially from using many sampling steps, whereas standard flow matching with Euler integration typically uses far fewer steps.
>
> (ii) **Flow vs. stochastic dynamics.** Flow matching learns the mean probability velocity field, while diffusion-based methods correspond to SDEs with an explicit diffusion term. This stochasticity can partially compensate for training noise and approximation error, which may explain why diffusion models perform more robustly than the deterministic base flow in this setting.
>
> [1] Huang, Zhilin, et al. "Protein-ligand interaction prior for binding-aware 3d molecule diffusion models." The Twelfth International Conference on Learning Representations. 2024.
>
> [2] Song, Yuxuan, et al. "Equivariant flow matching with hybrid probability transport for 3d molecule generation." Advances in Neural Information Processing Systems 36 (2023): 549-568.
>
> [3] Cremer, Julian, et al. "FLOWR: Flow Matching for Structure-Aware De Novo, Interaction-and Fragment-Based Ligand Generation." arXiv preprint arXiv:2504.10564 (2025).
>
> [4] Huang, Jie, and Daiheng Zhang. "Molform: Multi-modal flow matching for structure-based drug design." arXiv preprint arXiv:2507.05503 (2025).
>
> [5] Lee, Seul, Jaehyeong Jo, and Sung Ju Hwang. "Exploring chemical space with score-based out-of-distribution generation." International Conference on Machine Learning. PMLR, 2023.

---

### Meta-Review · Area_Chair_npT3 · 2025-12-20

**Summary:**

**Paper summary.** This paper proposes Perturbed Flow Matching (PFM) for pocket-conditioned ligand generation / structure-based drug design. The main idea is to modify the conditional probability path used in flow matching by adding a “perturbation” derived from protein-pocket and intra-ligand features, with the goal of improving generation quality and docking scores. The paper reports strong results on CrossDocked-style benchmarks.

**What happened in the discussion.** The core review concerns were: (1) the method is hard to understand (motivation and intuition are unclear), (2) novelty is unclear (may be an incremental modification of existing conditional paths / self-conditioning ideas), (3) baseline coverage is incomplete (missing other flow-matching SBDD methods and recent strong diffusion baselines), and (4) reporting is not strong enough (need error bars/CI, clarify guidance settings, and ensure fair speedup comparisons). In rebuttal, the authors did respond with concrete additions: they provided a clearer intuitive explanation, added comparisons to recent flow-matching baselines (e.g., MolFORM and FLOWR) with tables, and discussed additional baselines (MOOD) as future work. However, a reviewer follow-up comment still flagged unresolved issues: guidance notation and fairness (cfg/cg), missing strong baselines, and speedup claims being confounded by different sampling hyperparameters (NFE) rather than model/algorithm alone.

**My assessment as AC.** The direction is promising and the rebuttal improved baseline coverage. Still, after reading the discussion, I agree with reviewers that the current version is not yet solid enough: the paper’s key claims depend on careful, fair comparisons (including guidance settings and sampling steps), and the method should be explained in a simpler, more transparent way. At the moment, the story feels like it is still moving during rebuttal (adding baselines, clarifying evaluation), which makes it risky to accept in a highly competitive conference.

**Decision.** Reject. This is not a statement that the idea cannot work. I encourage the authors to (a) rewrite the method section with a clear, step-by-step algorithm description and intuition, (b) add strong and clearly configured baselines (including flow-matching and diffusion methods) with consistent sampling budgets and guidance settings, and (c) report uncertainty (error bars/CI) and disentangle speedup from changes in sampling hyperparameters. With those changes, the work could be more competitive in a future submission.

**Reviewer Concerns:**

- **Reviewer tvNa (rating 2, confidence 3)**:
  - Concerns: hard to read; unclear intuitive motivation; missing comparisons to other flow-matching methods (and additional diffusion baselines).
  - Author response: added a clearer intuition paragraph and added comparisons to flow-matching baselines (MolFORM and FLOWR) with tables; MOOD is discussed as future work.
  - Status: **partially resolved** (clarity improved, but baseline coverage and evaluation framing remain incomplete).
- **Reviewer C3Pp (rating 4, confidence 3)**:
  - Concerns: need statistical significance (error bars/CI), more visualizations (architecture and sample outputs), and comparison to another flow method (e.g., DrugFlow).
  - Author response: addressed uncertainty reporting and added additional comparisons; some items remain as future work.
  - Status: **partially resolved**.
- **Reviewer xeej (rating 2, confidence 4)**:
  - Concerns: novelty and theoretical benefit over diffusion/standard flow matching are not clear; fairness of comparisons when guidance and sampling hyperparameters differ; missing several strong recent baselines.
  - Author response: added more baseline comparisons and more discussion; however, the reviewer posted a follow-up comment still flagging missing baselines, guidance notation issues (cfg/cg), and that the speedup claim is confounded by different NFE/sampling settings.
  - Status: **not resolved** (the key fairness/novelty concerns remain).

**Reviewer Scores:**

- **Reviewer C3Pp (rating 4, confidence 3)**: Mildly positive but requests statistical significance and more visualizations; could remain borderline if addressed.
- **Reviewer xeej (rating 2, confidence 4)**: Negative on novelty and lack of theoretical insight; likely unchanged.
- **Reviewer tvNa (rating 2, confidence 3)**: Negative on clarity and missing flow-matching comparisons; likely unchanged.

---

### Decision · Program_Chairs · 2026-01-26

Reject